# A data-model approach to interpreting speleothem oxygen isotope records from monsoon regions

Sarah E. Parker[1], Sandy P. Harrison[1], Laia Comas-Bru[1], Nikita Kaushal[2], Allegra N. LeGrande[3], Martin Werner[4]

[1]School of Archaeology, Geography and Environmental Science, Reading University, Reading, UK
[2]Asian School of the Environment, Nanyang Technological University, Singapore
[3]NASA Goddard Institute for Space Studies and Center for Climate Systems Research, Columbia University, New York, USA
[4]Alfred Wegener Institute, Helmholtz Centre for Polar and Marine Research, Bremerhaven, Germany

*Correspondence to*: Sarah E. Parker (s.parker@pgr.reading.ac.uk)

**Abstract.** Reconstruction of past changes in monsoon climate from speleothem oxygen isotope ($\delta^{18}O$) records is complex because $\delta^{18}O$ signals can be influenced by multiple factors including changes in precipitation, precipitation recycling over land, temperature at the moisture source and changes in the moisture source region and transport pathway. Here, we analyse >150 speleothem records from version 2 of the Speleothem Isotopes Synthesis and Analysis (SISAL) database to produce composite regional trends in $\delta^{18}O$ in monsoon regions; compositing minimises the influence of site-specific karst and cave processes that can influence individual site records. We compare speleothem $\delta^{18}O$ observations with isotope-enabled climate model simulations to investigate the specific climatic factors causing these regional trends. We focus on differences in $\delta^{18}O$ signals between the mid-Holocene, the peak of the Last Interglacial (Stage 5e) and the Last Glacial Maximum, and on $\delta^{18}O$ evolution through the Holocene. Differences in speleothem $\delta^{18}O$ between the mid-Holocene and Last Interglacial in the East Asian and Indian monsoons are small, despite the larger summer insolation values during the Last Interglacial. Last Glacial Maximum $\delta^{18}O$ values are significantly less negative than interglacial values. Comparison with simulated glacial-interglacial $\delta^{18}O$ shows that changes are principally driven by global shifts in temperature and regional precipitation. Holocene speleothem $\delta^{18}O$ records show distinct and coherent regional trends. Trends are similar to summer insolation in India, China and southwestern South America, but different in the Indonesian-Australian region. Redundancy analysis shows that 37% of Holocene variability can be accounted for by latitude and longitude, supporting the differentiation of records into individual monsoon regions. Regression analysis of simulated precipitation $\delta^{18}O$ and climate variables show significant relationships between global Holocene monsoon $\delta^{18}O$ trends and changes in precipitation, atmospheric circulation and (to a lesser extent) source area temperature, whilst precipitation recycling is non-significant. However, there are differences in regional scale mechanisms; there are clear relationships between changes in precipitation and in $\delta^{18}O$ for India, southwestern South America and the Indonesian-Australian regions, but not for the East Asian monsoon. Changes in atmospheric circulation contributes to $\delta^{18}O$ trends in the East Asian, Indian and Indonesian-Australian monsoons, and a weak source area temperature effect is observed over southern and central America and Asia. Precipitation recycling is influential in southwestern South America

and southern Africa. Overall, our analyses show that it is possible to differentiate the impacts of specific climatic mechanisms influencing precipitation $\delta^{18}O$ and use this analysis to interpret changes in speleothem $\delta^{18}O$.

## 1 Introduction

The oxygen isotopic ($\delta^{18}O$: $^{18}O/^{16}O$ ratio relative to a standard, in permil, ‰) composition of speleothems is widely used to infer past regional climates (Bar-Matthews et al., 1997; McDermott, 2004; Wang et al., 2008). Speleothem oxygen isotope ($\delta^{18}O_{spel}$) signals are inherited from $\delta^{18}O$ in precipitation ($\delta^{18}O_{precip}$) above the cave, which in turn is determined by the initial $\delta^{18}O$ of water vapour as it evaporates at the oceanic moisture source region, the degree of rainout and evaporation from source to cave site and air temperature changes encountered throughout the moisture transport pathway (Fairchild and Baker, 2012; Lachniet, 2009). Understanding the effects and contribution of each of these climate processes to $\delta^{18}O_{precip}$ (and therefore $\delta^{18}O_{spel}$) is essential to inferring palaeoclimate from speleothem $\delta^{18}O$ records.

Initial $\delta^{18}O$ is determined by oceanic $\delta^{18}O$ at the evaporative moisture source region (Craig and Gordon, 1965), which varies spatially (LeGrande and Schmidt, 2006) and through time (e.g. Waelbroeck et al., 2002). During evaporation from the moisture source, $^{16}O$ is preferentially incorporated into the vapour, whilst subsequent fractionation during atmospheric transport occurs 45   by Rayleigh distillation; As air masses cool and moisture condenses, heavier $^{18}O$ is enriched in the liquid phase and removed by precipitation. With progressive rainout along a moisture pathway, precipitation becomes gradually more depleted (Dansgaard, 1964). Within this framework, $\delta^{18}O_{precip}$ is controlled by two variables: temperature and the amount of precipitation along a moisture pathway. The temperature effect stems from the cooling required for progressive rainout during Rayleigh distillation (Dansgaard, 1964; Rozanski et al., 1993). The temperature-$\delta^{18}O$ impact is dominant at mid to high 50   latitudes, whilst observations suggest that changes in upstream and local precipitation dominate changes in the $\delta^{18}O_{precip}$ signal at tropical latitudes. The negative relationship between local precipitation and $\delta^{18}O_{precip}$, often referred to as the "amount effect" (Bailey et al., 2018; Dansgaard, 1964), results from the re-evaporation and diffusive exchange between precipitation and water vapour during deep convective precipitation (Risi et al., 2008). However, Rayleigh distillation is complicated by changes in atmospheric circulation and moisture recycling. Changes in the area from which the moisture is sourced will modify $\delta^{18}O_{precip}$ 55   because the initial $\delta^{18}O$ values differ between sources (Cole et al., 1999; Friedman et al., 2002), whilst changes in the moisture transport pathway and/or distance between source and cave site can result in differing degrees of fractionation associated with condensation and evaporation (Aggarwal et al., 2012; Bailey et al., 2018). The isotopic composition of atmospheric water vapour may also be modified by precipitation recycling over land, since evapotranspiration returns moisture from precipitation back to the atmosphere thereby minimising the $\delta^{18}O_{precip}$/distance gradient along an advection path that occurs with Rayleigh 60   distillation (Gat, 1996; Salati et al., 1979).

Speleothem $\delta^{18}O$ records from monsoon regions show multi-millennial variability that has been interpreted as documenting the waxing and waning of the monsoons in response to changes in summer insolation, often interpreted predominantly as a

change in the absolute amount of precipitation (Cheng et al.,2013; Fleitmann et al., 2003) or a change in the ratio of more negative $\delta^{18}O$ summer precipitation to less negative $\delta^{18}O$ winter precipitation (Dong et al., 2010; Wang et al., 2001). However, the multiplicity of processes that influence $\delta^{18}O$ before incorporation in the speleothem make it difficult to attribute the climatic causes of changes in individual speleothem records unambiguously. In the East Asian monsoon, for example, speleothem $\delta^{18}O$ records have been interpreted as a summer monsoon signal, manifested as a change in the amount of water vapour removed along the moisture trajectory (Yuan et al., 2004), and/or as a change in the contribution of summer precipitation to annual totals (Cheng et al., 2006, 2009, 2016; Wang et al., 2001) based on the relationship between modern $\delta^{18}O_{precip}$ and climate. Other interpretations of Chinese monsoon $\delta^{18}O_{spel}$ have included rainfall source changes (Tan 2009, 2011, 2014) or local rainfall changes (Tan et al., 2015). Maher (2008) interpreted $\delta^{18}O_{spel}$ as reflecting changes in moisture source area, based on differences between $\delta^{18}O_{spel}$ and loess/palaeosol records of rainfall and the strong correlation between East Asian and Indian monsoon speleothems. Maher and Thompson (2012) used a mass balance approach to show that the changes in precipitation (either local or upstream) or rainfall seasonality required to reproduce $\delta^{18}O_{spel}$ trends would be unreasonably large. They therefore argued that changes in moisture source were required to explain shifts in $\delta^{18}O$ both on glacial/interglacial time scales and during interglacials. Overall, there are several plausible climate mechanisms that could contribute to $\delta^{18}O_{spel}$ on multi-millennial timescales. East Asian monsoon speleothem records are often interpreted as a combination of several of these processes (Cheng et al., 2016; Dykoski et al., 2005) which overall represent monsoon intensity (Cheng et al., 2019). There are also multiple interpretations of the causes of $\delta^{18}O_{spel}$ variability in other monsoon regions. In the Indian monsoon region, speleothem $\delta^{18}O$ records are interpreted primarily as an amount effect signal (Berkelhammer et al., 2010; Fleitmann et al., 2004), supported by $\delta^{18}O_{precip}$/climate observations (e.g. Battacharya et al., 2003). However, other studies have suggested that $\delta^{18}O_{precip}$ changes in this region are driven primarily by large-scale changes in monsoon circulation and hence, Indian monsoon $\delta^{18}O_{spel}$ should be interpreted as a moisture source/trajectory signal (Breitenbach et al., 2010; Sinha et al., 2015). In the Indonesian-Australian monsoon region, $\delta^{18}O_{spel}$ variability has been interpreted as a precipitation amount signal (Carolin et al., 2016; Krause et al., 2019) or a precipitation seasonality signal (Ayliffe et al., 2013; Griffiths et al., 2009), based on modern $\delta^{18}O_{precip}$ and climate observations (Cobb et al., 2007; Moerman et al., 2013), and/or as a moisture source/trajectory signal (Griffiths et al., 2009; Wurtzel et al., 2018). South American speleothem records have been interpreted as records of monsoon intensity, due to changes in the amount of precipitation over the region (Cruz et al., 2005; Wang et al., 2006; Cheng et al., 2013), changes in the degree of upstream precipitation and evapotranspiration (Cheng et al., 2013) or changes in the ratio of precipitation sourced from the low-level jet versus the Atlantic (Cruz et al., 2006; Wang et al., 2006).

These interpretations generally rely on modern $\delta^{18}O_{precip}$-climate observations, which may not have remained constant through time. The sources of $\delta^{18}O$ variability can also be explored using isotope-enabled climate models (e.g. Hu et al., 2019), which incorporate known isotope effects and therefore provide plausible explanations for $\delta^{18}O_{spel}$ trends. Modelling studies suggest that changes in East Asian monsoon $\delta^{18}O_{precip}$ (during Heinrich events: Lewis et al., 2010; Pausata et al., 2010, and on orbital timescale: Battisti et al., 2014; LeGrande and Schmidt, 2009) do not reflect local rainfall variability but instead reflect changes

in $\delta^{18}O$ of vapour delivered to the region. Variability in the $\delta^{18}O$ of vapour delivered to East Asia on orbital timescales has been diagnosed as due to changes in precipitation upstream of the region (Battisti et al., 2014), changes in moisture source location (Hu et al., 2019; Tabor et al., 2018) or changes in the strength of monsoon winds (LeGrande and Schmidt, 2009; Liu et al., 2014). $\delta^{18}O_{precip}$ variability in the East Asian monsoon during Heinrich events has also been attributed to non-local isotope fractionation (Lewis et al., 2010; Pausata et al., 2011). Modelling results suggest that changes in precipitation amount are the predominant source of $\delta^{18}O$ variability in the Indian monsoon during the Holocene (LeGrande and Schmidt, 2009) and in the glacial (Lewis et al., 2010), and in the South American and Indonesian/Australian regions during Heinrich events (Lewis et al., 2010) and the Last Interglacial (Sjolte and Hoffman, 2014).

In this study, we combine speleothem $\delta^{18}O$ records from version 2 of the Speleothem Isotopes Synthesis and Analysis (SISAL) database with isotope-enabled palaeoclimate simulations from two climate models to investigate the plausible mechanisms driving changes in $\delta^{18}O$ in monsoon regions through the Holocene (last 11,700 years) and between the mid-Holocene, the peak of the Last Interglacial and the Last Glacial Maximum. We compare $\delta^{18}O_{spel}$ signals across geographically separated cave sites to extract a regional signal, thus minimising the influence of karst and in-cave processes, such as the mixing of groundwaters from different precipitation events or changes in cave ventilation, that can be important for the $\delta^{18}O_{spel}$ of individual records. We use Principal Coordinate Analysis (PCoA) to identify regions with geographically coherent $\delta^{18}O_{spel}$ records, and then examine how these regions behave on glacial-interglacial time scales and through the Holocene. We use isotope-enabled model simulations, to investigate the potential causes of $\delta^{18}O_{spel}$ variability in regions where the models reproduce the large-scale $\delta^{18}O$ changes shown by observations. We exploit the fact that models produce internally physically consistent changes to explore potential and plausible causes of the trends observed in speleothem records across specific monsoon regions, using multiple regression analysis.

## 2. Methods

### 2.1 Speleothem oxygen isotope data

Speleothem $\delta^{18}O$ records were obtained from the SISAL (Speleothem Isotopes Synthesis and Analysis) database (Atsawawaranunt et al., 2018; Comas-Bru et al., 2020a, 2020b). Records were selected based on the following criteria:

- They are located in monsoon regions, between 35°S and 40°N;
- The mineralogy is known and does not vary (i.e. between calcite and aragonite) through time, because oxygen isotope fractionation during speleothem precipitation is different for calcite and aragonite;
- For the analysis of mid-Holocene (MH), Last Glacial Maximum (LGM) and Stage 5e during the Last Interglacial (LIG) $\delta^{18}O$ signals, the records contain samples within at least one of these time periods, defined as 6,000±500 years BP for the MH, 21,000±1,000 years BP for the LGM and 125,000±1,000 years BP for the LIG, where BP (before present) is 1950 CE;
- For the PCoA, the records have a temporal coverage of at least 4,000 years in the Holocene;

- For Holocene trend analyses, speleothems have a record of the period from 7,000 to 3,000 years BP;

- They are the most recent update of the record from a site available in version 2 of the SISAL database.

This resulted in the selection of 125 records from 44 sites for the PCoA analysis, 64 records from 38 sites for the analysis of MH, LGM and LIG signals and 79 records from 40 sites for the Holocene trend analysis (Fig. 1). Although the SISALv2 database contains multiple age models for some sites, we use the published age models given by the original authors for all records.

## 2.2 Climate model simulations

There are relatively few paleoclimate simulations made with models that incorporate oxygen isotope tracers, and the available simulations do not necessarily focus on the same periods or use the same modelling protocols. Here, we use simulations of opportunity from two isotope-enabled climate models: ECHAM5 (version 5 of the European Centre for medium range weather forecasting model in HAMburg) and GISS E-R (Goddard Institute for Space Studies Model version E-R). The ECHAM5 simulations provide an opportunity to examine large-scale changes between glacial and interglacial states, using simulations

of the MH, LGM and LIG. The GISS Model E-R Ocean-Atmosphere Coupled General Circulation Model was used to investigate the evolution of $\delta^{18}O$ evolution during the Holocene, using eight time slices (9 ka, 6 ka, 5 ka, 4 ka, 3 ka, 2 ka, 1 ka and 0 ka) experiments. Although simulations of the MH 6ka time slice are available with both models, there are differences in the protocol used for the two experiments which preclude direct comparison of the simulations.

The ECHAM5-wiso MH experiment (Wackerbarth et al., 2012; Werner, 2019) was forced by orbital parameters (based on

Berger and Loutre, 1991) and greenhouse gas (GHG) concentrations ($CO_2$ = 280 ppm, $CH_4$ = 650 ppb, $N_2O$ = 270 ppb) appropriate to 6 ka. Changes in sea-surface temperature (SST) and sea-ice were derived from a transient Holocene simulation (Varma et al., 2012). The control simulation for the MH experiment was an ECHAM-wiso simulation of the period 1956-1999 (Langebroek et al., 2011), using observed SSTs and sea-ice cover. This control experiment was forced by SSTs and sea-ice only, with atmospheric circulation free to evolve. The ECHAM5-wiso LGM experiment (Werner, 2019; Werner et al., 2018)

was forced by orbital parameters (Berger and Loutre, 1991), GHG concentrations ($CO_2$ = 185 ppm, $CH_4$ = 350 ppb, $N_2O$ = 200 ppb), land-sea distribution and ice sheet height and extent appropriate to 21 ka; SST and sea-ice cover were prescribed from the GLAMAP dataset (Schäfer-Neth and Paul, 2003). Sea surface water and sea-ice $\delta^{18}O$ were uniformly enriched by 1 ‰ at the start of the experiment. The control simulation for the LGM experiment used present-day conditions, including orbital parameters and GHG concentrations set to modern values, and SSTs and sea-ice cover from the last 20 years (1979-1999).

Both the MH and LGM simulations were run at T106 horizontal grid resolution, approximately 1.1° by 1.1°. Comparison of the MH and LGM simulations with speleothem data globally (Comas-Bru et al., 2019; Fig. S1 and S2) show that the ECHAM model reproduces the broadscale spatial gradients and the sign of isotopic changes at the majority of cave sites (MH: 72%; LGM: 76%). However, the changes compared to present are generally more muted in the simulations than shown by the speleothem records.

The LIG experiment (Gierz et al., 2017b, 2017a) was run using the ECHAM5/MPI-OM Earth System Model, with stable water isotope diagnostics included in the ECHAM5 atmosphere model (Werner et al., 2011), the dynamic vegetation model JSBACH (Haese et al., 2012) and the MPI-OM ocean/sea-ice module (Xu et al., 2012). This simulation was run at a T31L19 horizontal grid resolution, approximately 3.75° by 3.75°. The LIG simulation was forced by orbital parameters derived from Berger and Loutre (1991) and GHG concentrations ($CO_2$ = 276 ppm, $CH_4$ = 640 ppb, $N_2O$ = 263 ppb) appropriate to 125 ka, but it was

assumed that ice sheet configuration and land-sea geography is unchanged from modern and therefore no change was made to the isotopic composition of sea water. The LIG simulation is compared to a pre-industrial (PI) control with appropriate insolation, GHG and ice sheet forcing for 1850 CE. The sign of simulated isotopic changes in the LIG is in good agreement with ice core records from Antarctica and Greenland and speleothem records from Europe, the Middle East and China (Gierz et al., 2017b) although, as with the MH and LGM, the observed changes tend to be larger than the simulated changes (Fig. S3).

There are GISS ModelE-R (LeGrande and Schmidt, 2009) simulations for eight time slices during the Holocene (9 ka, 6 ka, 5 ka, 4 ka, 3 ka, 2 ka, 1 ka and 0 ka). The 0 ka experiment is considered as the pre-industrial control (ca 1880 CE). Orbital parameters were based on Berger and Loutre (1991) and GHG concentrations were adjusted based on ice core reconstructions (Brook et al., 2000; Indermühle et al., 1999; Sowers, 2003) for each time slice. A remnant Laurentide ice sheet was included in the 9 ka simulation, following Licciardi et al. (1998), and the corresponding adjustment was made to mean ocean salinity

and ocean water $\delta^{18}O$ to account for this (Carlson et al., 2008). The ice sheet in all the other experiments was specified to be the same as modern, and therefore no adjustment was necessary. The simulations were run using the M20 version of GISS ModelE-R, which has a horizontal resolution of 4° by 5°. Each experiment was run for 500 years and we use the last 100 simulated years for the analyses. Comparison of the simulated trends in $\delta^{18}O$ show good agreement with Greenland ice core records, marine records from the tropical Pacific and Chinese speleothem records (LeGrande and Schmidt, 2009). However,

as is the case with the ECHAM simulations, the model tends to produce changes less extreme than shown by the observations (Fig S4, S5 and S6).

## 2.3 Principal Coordinate Analysis and Redundancy Analysis

We used PCoA to identify regionally coherent patterns in the speleothem $\delta^{18}O$ records for the Holocene. PCoA is a multivariate ordination technique that uses a distance/dissimilarity matrix to represent inter-object (dis)similarity in reduced space (Gower,

1966; Legendre and Legendre, 1998). Speleothem records from individual sites are often discontinuous; missing data is problematic for many ordination techniques. PCoA is more robust to missing data than other methods (Kärkkäinen and Saarela, 2015; Rohlf, 1972). We used a correlation matrix of speleothem records as the (dis)similarity measure. The temporal resolution of speleothem records was first standardised by calculating a running average mean with non-overlapping 500-year windows. This procedure produces a single composite record when there are several records for a given site. PCoA results

were displayed as a biplot, where sites ordinated close to one another (i.e., with similar PCoA scores) show similar Holocene trends and sites ordinated far apart have dissimilar trends. We used the 'broken stick' model (Bennett, 1996) to identify which

PCoA axes were significant. We used redundancy analysis (RDA: Legendre and Legendre, 1998; Rao, 1964) with latitude and longitude as predictor variables to identify if PCoA (dis)similarities were related to geographical location, and Principal Components Analysis (PCA) to identify the main patterns of variation. As these explanatory variables are not dimensionally homogeneous, they were centred on their means and standardised to allow direct comparison of the gradients. PCoA and RDA analyses were carried out using the 'vegan' package in R (Oksanen et al., 2019).

## 2.4 Glacial-interglacial changes in $\delta^{18}O$

We examined shifts in $\delta^{18}O_{spel}$ observations and in annual precipitation-weighted mean $\delta^{18}O_{precip}$ from ECHAM-wiso in regions influenced by the monsoon, between the MH, LGM and LIG. Values are given as anomalies with respect to the present-day for speleothems or the control simulation experiment for model outputs. Comas-Bru et al. (2019) have shown that differences in speleothem $\delta^{18}O$ data between the 20th century and the pre-industrial period (i.e. 1850±15 CE) are within the temporal and measurement uncertainties of the data, and thus the use of different reference periods (i.e. PI for the ECHAM LIG experiment, 20th century for ECHAM MH, LGM experiments) should have little effect on our analyses. We used mean site $\delta^{18}O_{spel}$ values for each period for the regions identified in the PCoA analysis. Where there are multiple speleothem $\delta^{18}O$ records for a site in a time period, they were averaged to calculate mean $\delta^{18}O_{spel}$. Three sites above 3500m were excluded from the calculation of the means because high elevation sites have more negative $\delta^{18}O$ values than their low-elevation counterparts and their inclusion would distort the regional estimates.

There are relatively few speleothems covering both the present-day and the period of interest (i.e., MH, LGM or LIG), precluding the calculation of $\delta^{18}O_{spel}$ anomalies from the speleothem data. We therefore calculated anomalies with respect to modern (1960-2017 CE) using as reference the Online Isotopes in Precipitation Calculator (OIPC: Bowen, 2018; Bowen and Revenaugh, 2003), a global gridded dataset of interpolated mean annual precipitation-weighted $\delta^{18}O_{precip}$ data. This dataset combines data from 348 stations from the Global Network of Isotopes in Precipitation (IAEA/WMO, 2018), covering part or all of the period 1960-2014, and other records available at the Water Isotopes Database (Waterisotopes Database, 2017). OIPC $\delta^{18}O_{precip}$ was converted to its speleothem equivalent assuming that: (i) precipitation-weighted mean annual $\delta^{18}O_{precip}$ is equivalent to mean annual drip-water $\delta^{18}O$ (Yonge et al., 1985) and (ii) precipitation of calcite is consistent with the empirical speleothem-based kinetic fractionation factor of Tremaine et al. (2011) and precipitation of aragonite follows the fractionation factor from Grossman and Ku (1986), as formulated by Lachniet (2015):

$$\delta^{18}O_{calcite\_SMOW} = w\delta^{18}O_{precip\_SMOW} + \left(\left(\frac{16.1 \cdot 1000}{T}\right) - 24.6\right) \qquad \text{(T in K)} \qquad (1)$$

$$\delta^{18}O_{aragonite\_SMOW} = w\delta^{18}O_{precip\_SMOW} + \left(\left(\frac{18.34 \cdot 1000}{T}\right) - 31.954\right) \qquad \text{(T in K)} \qquad (2)$$

where $\delta^{18}O_{calcite\_SMOW}$ and $\delta^{18}O_{aragonite\_SMOW}$ are the speleothem isotopic composition for calcite and aragonite speleothems with reference to the V-SMOW standard (in permil); $w\delta^{18}O_{precip}$ is the OIPC precipitation-weighted annual mean isotopic

composition of precipitation with respect to the V-SMOW standard and T is the mean annual cave temperature (in degrees Kelvin). We used the long-term (1960-2016) mean annual surface air temperature from the CRU-TS4.01 database (Harris et al., 2014) at each site as a surrogate for mean annual cave air temperature. The resolution of the gridded data means that $w\delta^{18}O_{precip\_SMOW}$ and T may be the same for nearby sites.

We use the V-SMOW to V-PDB conversion from Coplen et al. (1983), which is independent of speleothem mineralogy:

$$\delta^{18}O_{PDB} = 0.97001 \cdot \delta^{18}O_{SMOW} - 29.29 \tag{3}$$

where $\delta^{18}O_{PDB}$ is relative to the V-PDB standard and $\delta^{18}O_{SMOW}$ is relative to V-SMOW standard.

Average uncertainties in the speleothem age-depth models are ~50 years during the Holocene. This interval is smaller than the time windows used in this analysis, and age uncertainty is therefore expected to have a negligible impact on the results. We investigated the influence of age uncertainties on the LGM and LIG $\delta^{18}O_{spel}$ anomalies by examining the impact of using different window widths ($\pm 500$, $\pm 700$, $\pm 1000$, $\pm 2000$ years) on the regional mean $\delta^{18}O_{spel}$ anomalies.

We used anomalies of $w\delta^{18}O_{precip}$, mean annual surface air temperature (MAT) and mean annual precipitation (MAP) from the ECHAM5-wiso simulations to investigate the changes in $\delta^{18}O_{spel}$ between the MH, LGM and LIG, and their association with changes in climate. Values were calculated from land grid cells (>50% land) $\pm 3°$ around each speleothem site. This distance was chosen with reference to the coarsest resolution simulation (LIG, ca. 3.75 x 3.75°). Gridded values of MAT and MAP were weighted by the proportion of each grid cell that lies within $\pm 3°$ of the site and linear distance-weighted means were calculated for each site and time slice. We only considered regions with at least one speleothem record for each of the three time periods, although these were not required to be the same sites, and where the observed shifts in $\delta^{18}O_{spel}$ were in the same direction and of a similar magnitude to the simulated $w\delta^{18}O_{precip}$.

## 2.5 Holocene and Last Interglacial regional trends

Regional speleothem $\delta^{18}O$ changes through the Holocene were examined by creating composite time-series for each region identified in the PCoA analysis with at least four Holocene records (> 5000 years long). Regional composites were constructed using a 4-step procedure, modified from Marlon et al. (2008): (i) the $\delta^{18}O$ data for individual speleothems were transformed to z-scores, so all records have a standardised mean and variance:

$$z\text{-}score_i = \left(\delta^{18}O_i - \overline{\delta^{18}O}_{(base\ period)}\right)/s\delta^{18}O_{(base\ period)} \tag{4}$$

Where $\overline{\delta^{18}O}$ is the mean and $s\delta^{18}O$ is the standard deviation of $\delta^{18}O$ for a common base period. A base period of 7,000 to 3,000 years BP was chosen to maximise the number of records included in each composite. (ii) the standardised data for a site were re-sampled by applying a 100-year non-overlapping running mean with the first bin centred at 50 years BP, in order to create a single site time series while ensuring that highly resolved records do not dominate the regional composite; (iii) each regional composite was constructed using locally weighted regression (Cleveland and Devlin, 1988) with a window width of 3,000 years and fixed target points in time; and (iv) confidence intervals (5th and 95th percentiles) for each composite were generated by bootstrap resampling by site over 1,000 iterations. There are too few sites to construct regional composites for

the peak of the LIG (Stage 5e) and thus the trends in $\delta^{18}O_{spel}$ were examined using records from individual sites covering the period 130-116 ka BP.

We calculated Holocene regional composites from annual precipitation-weighted mean $\delta^{18}O_{precip}$ anomalies simulated by the GISS model. Simulated $\delta^{18}O_{precip}$ trends were calculated using linear distance-weighted mean $\delta^{18}O_{precip}$ values from land grid cells (>50% land) within ±4° around each site. This distance was determined by the grid resolution of the model. Regional
composites were then produced using bootstrap resampling in the same way as for the speleothem data. The simulated anomalies are relative to the control run rather than the specified base period used for the speleothem-based composites, so absolute values of simulated and observed Holocene trends are expected to differ. Preliminary analyses showed that neither the mean values nor trends in $\delta^{18}O_{precip}$ were substantially different if the sampled area was reduced to match the sampling used for the ECHAM-based box plot analysis, or was increased to encompass the larger regions shown in Fig. 1 and used in
the multiple regression analysis.

## 2.6 Multiple regression analysis

We investigate the underlying relationships between regional $\delta^{18}O_{precip}$ (and by extension $\delta^{18}O_{spel)}$) and monsoon climate through the Holocene using multiple linear regression (MLR). We use annual precipitation-weighted mean $\delta^{18}O_{precip}$ anomalies and climate variables from GISS model E-R. Climate variables were chosen to represent the four potential large-scale drivers of
regional changes in the speleothem $\delta^{18}O$ records. Specifically, we use changes in mean precipitation and precipitation recycling over the monsoon regions, and changes in mean surface air temperature and surface wind direction over the moisture source regions. Whereas the influence of changes in precipitation, recycling and temperature are relatively direct measures, the change in surface wind direction over the moisture source region is used as an index of potential changes in the moisture source region and transport pathway. The boundaries of each monsoon region (Fig. 1) were defined to include all the speleothem sites used
to construct the Holocene $\delta^{18}O_{spel}$ composites. Moisture source area limits (Fig. 1) were defined based on moisture tracking studies (Bin et al., 2013; Breitenbach et al., 2010; D'Abreton and Tyson, 1996; Drumond et al., 2008, 2010; Durán-Quesada et al., 2010; Kennett et al., 2012; Nivet et al., 2018; Wurtzel et al., 2018) and GISS simulated summer surface winds. All climate variables were extracted for the summer months, defined as May to September (MJJAS) for northern hemisphere regions and November to March (NDJFM) for southern hemisphere regions (Wang and Ding, 2008) on the basis that these
regions are dominated by summer season precipitation (Fig. S7). Only grid cells with >50% land were used to extract variables over monsoon regions and only grid cells with <50% land were used to extract variables over moisture source regions. The inputs to the MLR for each time interval were calculated as anomalies from the control run.

Precipitation recycling was calculated as the ratio of locally sourced precipitation versus total precipitation. Although the GISS E-R mid-Holocene experiment explicitly estimates recycling using vapour source distribution tracers (Lewis et al., 2014), this
was not done for all the Holocene time slice simulations. Therefore, we calculate a precipitation recycling index (RI), following Brubaker et al. (1993):

$$RI \; = \; \frac{P_R}{P} = \frac{E}{2Q_H + E} \tag{5}$$

Where locally sourced (recycled) precipitation ($P_R$) is estimated using total evaporation over a region (E) and total precipitation (P) is estimated as the sum of total evaporation and net incoming moisture flux integrated across the boundaries of the region (Q_H). RI therefore expresses the change in the contribution of local, recycled precipitation independently of any overall change in precipitation amount.

We incorporate mean meteorological variables and $\delta^{18}O_{precip}$ for all Holocene time slices (1ka to 9ka) and all monsoon regions into the MLR model. Thus, the relationships constrained by the overall (global) MLR model represent the combined response across all monsoon regions. We use pseudo-$R^2$ to determine the goodness-of-fit for the global MLR model, and t values (the regression coefficient divided by its standard error) to determine the strength of each relationship. Partial residual plots were used to show the relationship between each predictor variable and $\delta^{18}O_{precip}$ when the effects of the other variables are held constant.

All statistical analyses were performed in R (R Core Team, 2019) and plots were generated using ggplot (Wickham, 2016).

## 3 Results

### 3.1 Principal Coordinate Analysis and Redundancy Analysis

PCoA shows the (dis)similarity of Holocene $\delta^{18}O_{spel}$ evolution across individual records, and thus allows an objective regionalisation of these records. The first two PCoA axes are significant, according to the broken stick test, and account for 65% and 20% of $\delta^{18}O_{spel}$ variability respectively (Table 1). The PCoA scores differentiate records geographically (Fig. 2a): southern hemisphere monsoon regions such as the southwestern South American Monsoon (SW-SAM) and South African Monsoon (SAfM) are characterised by low PCoA1 scores, whilst northern hemisphere monsoons such as the Indian Summer Monsoon (ISM) and the East Asian Monsoon (EAM), are characterised by higher PCoA1 scores. This indicates that regions can be differentiated based on their temporal evolution as captured by the first PCoA axis. Most southern hemisphere regions also have lower PCoA2 scores although this is not consistent over time. Speleothem records from Central America (CAM) and Indonesian-Australian monsoon (IAM) have PCoA scores intermediate between the northern and southern hemisphere regions. PCoA clearly separates the South American records into a northeastern region (NE-SAM) with scores similar to other northern hemisphere monsoon regions and a southwestern region (SW-SAM), with scores similar to other southern hemisphere regions. The RDA supports a geographical control on the (dis)similarity of speleothem $\delta^{18}O$ records over the Holocene (Fig. 2b). RDA1 explains 37% of the variability and is significantly correlated with both latitude and longitude (Table 2).

### 3.2 Regional interglacial-glacial differences

To investigate the causes of shifts in $\delta^{18}O$ between the MH, LGM and LIG, we compare simulated and observed regional $\delta^{18}O$ signals during these periods with shifts in climate variables (precipitation and temperature). Only the ISM, EAM and IAM

regions have sufficient speleothem data (i.e. at least one record from every time period) to allow comparisons across the MH, LGM and LIG (Fig. 3) and have similar shifts in observed $\delta^{18}O_{spel}$ and simulated $\delta^{18}O_{precip}$. The regional mean $\delta^{18}O_{spel}$ anomalies calculated for different time windows ($\pm$ 500, $\pm$ 700, $\pm$ 1000, $\pm$ 2000 years) vary by less than 0.35‰ for the LGM (ISM: <0.16‰, EAM: <0.35‰, IAM: <0.22‰) and 0.48 ‰ for the LIG (ISM: <0.16‰, EAM: <0.48‰, IAM: <0.11‰), indicating that age uncertainties have a minimal impact on these mean values. The most positive $\delta^{18}O_{spel}$ anomalies in all three regions occur at the LGM, with more negative anomalies for the MH and LIG. The simulated $\delta^{18}O_{precip}$ anomalies show a similar pattern, with more positive anomalies during the LGM than during the MH or the LIG. The amplitude of this pattern is also similar between $\delta^{18}O_{precip}$ and $\delta^{18}O_{spel}$, when the observations are converted to their drip water equivalent (Fig. S8). The differences in regional $\delta^{18}O_{spel}$ anomalies between MH and LIG differ across the three regions. In both the ISM and the EAM, differences in $\delta^{18}O_{spel}$ values between the MH and LIG are small (Fig. 3a, 3b), although ISM LIG $\delta^{18}O_{spel}$ values are slightly more negative than MH values. In the IAM, MH values are less negative than the LIG (Fig. 3c). However, there are only a limited number of speleothem records from the ISM and IAM during the LIG, so the apparent differences between the two intervals in these regions may not be meaningful. Glacial-interglacial shifts are also seen in simulated temperature and precipitation, with warmer and wetter conditions during interglacials and cooler and drier conditions during the LGM in all three regions. Differences in simulated precipitation between the MH and the LIG could help explain the differences between $\delta^{18}O_{spel}$ in the ISM and IAM, since the LIG is wetter than the MH in the ISM and drier than the MH in the IAM. However, the LIG is also drier than the MH in the EAM, a feature that appears inconsistent with the lack of differentiation between the $\delta^{18}O$ signals in this region.

## 3.3 Regional-scale interglacial $\delta^{18}O$ evolution

There are four monsoon regions with sufficient data to examine regional Holocene $\delta^{18}O$ trends: EAM, ISM, SW-SAM and IAM (Fig. 4). The IAM region has the fewest records (n=7) whilst the EAM has the largest number (n=14). The regional composites are expressed as z-scores, i.e. changes with respect to the mean and variance of $\delta^{18}O$ for the base period (3000-7000 yr BP). The confidence intervals on the regional composites are small for all regions, except SW-SAM in the early Holocene. The EAM and ISM regions (Fig. 4 a-e) show the most positive $\delta^{18}O_{spel}$ z-scores around 12 ka followed by a rapid decrease towards their most negative values at ~9.5 ka and ~9 ka, respectively. The $\delta^{18}O_{spel}$ z-scores in the EAM are relatively constant from 9.5 to ~7 ka, whereas this plateau is present but less marked in the ISM. There is a gradual trend towards more positive $\delta^{18}O_{spel}$ z-scores towards the present in both regions thereafter. The SW-SAM records (Fig. 4i) have their most positive $\delta^{18}O_{spel}$ z-scores in the early Holocene with a gradual trend to more negative scores towards the present. By contrast, the IAM z-scores (Fig. 4g) are most positive at 12ka, gradually decreasing until ca 5 ka and are relatively flat thereafter.

There are insufficient data to create composite curves for the LIG, but individual records from the four regions (Fig. 5) show similar features to the Holocene trends. Records from the ISM and EAM (Fig. 5 left), for example, are characterised by an initial sharp decrease in $\delta^{18}O_{spel}$ values of about 4 ‰ between 130-129 ka and then most of the records (Dykoski et al., 2005;

Kathayat et al., 2016; Wang et al., 2008) show little variability for several thousand years. Despite the fact that the Tianmen record (Cai et al., 2010, 2012) shows considerable variability between 123-127 ka, there is nevertheless a similar plateau in the average observed value before the rapid change to less negative values after 127 ka. Similar to the Holocene, the SW-SAM record (Cheng et al., 2013) shows increasingly negative $\delta^{18}O_{spel}$ values through the LIG. The trend shown for Whiterock cave (Carolin et al., 2016) also shows similar features to the IAM Holocene composite, with a gradual trend towards more negative

values initially and a relatively complacent curve towards the end of the interglacial (Fig. 5 right).

### 3.4 Multiple regression analysis of Holocene $\delta^{18}O_{precip}$

    The MLR analyses of simulated $\delta^{18}O_{precip}$ trends identify the impact of an individual climate variable on $\delta^{18}O_{precip}$ in the absence of changes in other variables. The global MLR model includes the Holocene (1 to 9ka) $\delta^{18}O_{precip}$ trends combined across all monsoon regions (CAM, ISM, EAM, SW-SAM, NE-SAM, SAfM, IAM). This global monsoon MLR model has a pseudo-$R^2$

of 0.80 and shows statistically significant relationships between the anomalies in $\delta^{18}O_{precip}$ and anomalies in regional precipitation, temperature and surface wind direction (Table 3). The partial residual plots (Fig. 6) show there is a strong negative relationship with regional precipitation (t value = -8.75) and a strong positive relationship (t value = 8.03) with surface wind direction over the moisture source region, an index of changes in either source area or moisture pathway. This indicates that increases in regional precipitation alone will lead to a decrease in $\delta^{18}O$ while changes in source area/moisture pathway, in

the absence of changes in other variables, will lead to significant change in $\delta^{18}O$. The relationship with temperature over the moisture source region is weaker, but positive (t value = 2.05), i.e. an increase in temperature over the moisture source region will lead to an increase in $\delta^{18}O$ if there are no changes in other climate variables. Precipitation recycling is not significant in this global analysis. The exact choice of source region has a negligible impact on the model, for example expanding the ISM source region to include the Bay of Bengal does not change the outcome of this analysis (Fig S9, Table S1).

There are too few data points to make regressions for individual monsoon regions, but the distribution of data points for each region in the partial residual plots (Fig. 6) is indicative of the degree of conformity to the global MLR model (representing the combined response across all monsoon regions). Data points from the ISM, SW-SAM, IAM and SAfM are well aligned with the overall relationship with regional precipitation (Fig. 6a), indicating that precipitation is an important control on changes in $\delta^{18}O_{precip}$ in these regions. The NE-SAM, EAM and CAM values deviate somewhat from the overall relationship and, although

there are relatively few points, this suggests that changes in precipitation are a less important influence on $\delta^{18}O_{precip}$ changes in these regions. The impact of temperature changes (Fig. 6b) in the ISM, EAM and SW-SAM is broadly consistent with the overall relationship. The slope of the relationship with temperature is negative for the IAM and NE-SAM, and since this is physically implausible it suggests that some factor not currently included in the MLR is influencing these records. However, the inconsistencies between the regional signals helps to explain why the global relationship between anomalies in temperature

and $\delta^{18}O_{precip}$ is weak (Fig. 6b) and probably reflects the fact that tropical temperature changes during the Holocene are small. Data points from the EAM, ISM and IAM are well aligned with the overall relationship between changes in $\delta^{18}O_{precip}$ and

changes in wind direction (Fig. 6c), indicating that changes in source area or moisture pathway are an important control on changes in $\delta^{18}O_{precip}$ in these regions. However, values for CAM, SW-SAM, NE-SAM and SAfM deviate strongly from the overall relationship. Recycling does not appear to be an important contributor to changes in $\delta^{18}O_{precip}$ except in SW-SAM and
SAfM (Fig. 6d).

## 4 Discussion

We have shown that it is possible to derive an objective regionalisation of speleothem records based on PCoA of the oxygen-isotope trends through the Holocene (Fig. 2). This approach separates out regions with a distinctive northern hemisphere signal (e.g. ISM, EAM, NE-SAM) from regions with a distinctive southern hemisphere signal (e.g. SW-SAM, SAfM), reflecting the
fact that the evolution of regional monsoons in each hemisphere follows, to some extent, insolation forcing. It also identifies regions that have an intermediate pattern (e.g. IAM). The robustness of the regionalisation is borne out by the fact that Holocene composite trends from each region have tight confidence intervals (Fig. 4), showing that the signals of individual records across a region show broad similarities. The monsoon regions identified by PCoA are consistent with previous studies (Wang et al., 2014). The tracking of northern hemisphere insolation is a recognised feature of monsoon systems in India and China
(see reviews by Kaushal et al., 2018; Zhang et al., 2019). The separation of speleothem records from NE-SAM from those in SW-SAM is consistent with the precipitation dipole that exists between northeastern Brazil (Nordeste) and the continental interior (Berbery and Barros, 2002; Boers et al., 2014). The anti-phasing of speleothem records from the two regions during the Holocene has been recognised in previous studies (Cruz et al., 2009; Deininger et al., 2019). The intermediate nature of the records from the maritime continent is consistent with the fact that the Indonesian-Australian (IAM) summer monsoon is
influenced by cross-equatorial air flow and hence can be influenced by northern hemisphere conditions (Trenberth et al., 2000). Palaeoenvironmental records from this region show mixed signals for the Holocene: some have been interpreted as showing enhanced (Beaufort et al., 2010; Mohtadi et al., 2011; Quigley et al., 2010; Wyrwoll and Miller, 2001) and others reduced precipitation (Kuhnt et al., 2015; Steinke et al., 2014) during the early and mid-Holocene. Modelling studies have shown that this region is highly sensitive to SST changes in the Indian Ocean and South China Sea, which in turn reflect changes in the
northern hemisphere winter monsoons. Although most climate models produce a reduction in precipitation across the IAM during the mid-Holocene in response to orbital forcing, this is less than might be expected in the absence of ocean feedbacks associated with changes in the Indian Ocean (Zhao and Harrison, 2012).

The separation of northern and southern monsoon regions is consistent with the idea that changes in monsoon rainfall are primarily driven by changes in insolation (Ding and Chan, 2005; Kutzbach et al., 2008). Indeed, regional $\delta^{18}O_{spel}$ composites
from the EAM, ISM and SW-SAM show a clear relationship with the long-term trends in local summer insolation (Fig. 4). Similar patterns are seen in individual speleothem records from each region confirming that the composite trends are representative. However, the composite trends are not an exact mirror of the insolation signal over the Holocene. For example, the ISM and EAM composites show a more rapid rise during the early Holocene than implied by the insolation forcing. The

maximum wet phase in these two regions lasts for ca 3,000 years, again contrasting with the gradual decline in insolation

forcing after its peak at ca 11 ka. Both the rapid increase and the persistence of wet conditions for several thousand years is also observed in other palaeohydrological records across southern and central China, including pollen (Zhao et al., 2009; Li et al., 2018) and peat records (Hong et al., 2003; Zhou et al., 2004). These features are also characteristic of lake records from India (Misra et al., 2019). The lagged response to increasing insolation is thought to be due to the presence of northern hemisphere ice sheets in the early Holocene (Zhang et al., 2018). The persistence of wetter conditions through the early and

mid-Holocene is thought to reflect the importance of land-surface and ocean feedbacks in sustaining regional monsoons (Dallmeyer et al., 2010; Kutzbach et al., 1996; Marzin and Braconnot, 2009; Rachmayani et al., 2015; Zhao and Harrison, 2012). The evolution of regional monsoons during the LIG shows patterns similar to those observed during the Holocene, including the lagged response to insolation and the persistence of wet conditions after peak insolation. This is again consistent with the idea that internal feedbacks play a role in modulating the monsoon response to insolation forcing. We have also shown

that there is little difference in the isotopic values between the MH and the LIG in the ISM and EAM regions, which is also observed in individual speleothem records (Kathayat et al., 2016; Wang et al., 2008). The LIG (125ka) period was characterised by higher summer insolation, higher $CO_2$ concentrations (Otto-Bliesner et al., 2017) and lower ice volumes (Dutton and Lambeck, 2012) than the MH, suggesting that the LIG ISM and EAM monsoons should be stronger than the MH monsoons. The lack of a clear differentiation in the isotope signals between the LIG and MH suggests that other factors play a role in

modulating the monsoon response to these forcings and may reflect the importance of global constraints on the externally-forced expansion of the tropical circulation (Biasutti et al., 2018).

Global relationships between $\delta^{18}O_{precip}$ and climate variables (precipitation amount, temperature and surface wind direction; Fig. 6) are consistent with existing studies: a strong relationship with precipitation and a weaker temperature effect has been widely observed at tropical and sub-tropical latitudes in modern observations (Dansgaard, 1964; Rozanski et al., 1993). The

significant global relationship between $\delta^{18}O_{precip}$ and surface winds supports the idea that changes in moisture source and pathway are also important for explaining $\delta^{18}O$ variability over the Holocene. The multiple regression analysis also provides insights into the relative importance of different influences at a regional scale. In the ISM, the results support existing speleothem studies that suggest changes in precipitation amount (Cai et al., 2015; Fleitmann et al., 2004) and to a lesser extent moisture pathway (Breitenbach et al., 2010) drive $\delta^{18}O_{spel}$ variability. The $\delta^{18}O$ variability in the IAM region through the

Holocene also appears to be strongly driven by changes in precipitation and moisture pathway, consistent with the interpretation of Wurtzel et al. (2018). Changes in regional precipitation (where the cave sites are located) do not seem to explain the observed changes in $\delta^{18}O_{spel}$ in the EAM during the Holocene, where Holocene $\delta^{18}O_{precip}$ evolution is largely driven by changes in atmospheric circulation (indexed by changes in surface winds). This is consistent with existing studies that emphasise changes in moisture source and/or pathway rather than local precipitation changes (Maher, 2016; Maher and

Thompson, 2012; Tan, 2014; Yang et al., 2014). Speleothem $\delta^{18}O$ records in the SW-SAM clearly reflect regional-scale changes in precipitation, consistent with interpretations of individual records (Cruz et al., 2009; Kanner et al., 2013). However,

this is a region where changes in precipitation recycling also appear to be important. Based on regional water budget estimates, recycling presently contributes ca 25-35% of the precipitation over the Amazon (Brubaker et al., 1993; Eltahir and Bras, 1994), while these figures increase up to ca 40-60% based on moisture tagging studies (Risi et al., 2013; Yoshimura et al., 2004).

The LGM is characterised by lower northern hemisphere summer insolation, globally cooler temperatures, expanded global ice volumes and lower GHG concentrations than either the MH or the LIG. The MH and LIG (Stage 5e) periods represent peaks in the present and last interglacial periods, whilst the LGM represents maximum ice extent during the Last Glacial Period. Hence, comparison of these time periods provides a snap-shot view of glacial-interglacial variability. The $\delta^{18}O_{spel}$ anomalies are more positive during the LGM than the MH or LIG, suggesting drier conditions in the ISM, EAM and IAM,

supported by simulated changes in $\delta^{18}O_{precip}$ and precipitation (Fig. 3). Cooler SSTs of approximately 2°C (relative to the MH and LIG) in the ISM and EAM and of approximately 3°C in IAM source areas, together with a ca 5% decrease in relative humidity (Yue et al., 2011) would result in a water vapour $\delta^{18}O$ signal at the source ca 1 ‰ more depleted than seawater. This depletion results from the temperature dependence of equilibrium fractionation during evaporation and kinetic isotope effects related to humidity (Clark and Fritz, 1997). This fractionation counteracts any impact from enriched seawater $\delta^{18}O$ values

during the LGM (ca. +1 ‰ relative to the MH or LIG; Waelbroeck et al., 2002). Cooler air temperatures will also result in a depletion of $\delta^{18}O_{spel}$ during the LGM of ca 0.4 ‰ and 0.6 ‰ for the ISM/EAM and IAM respectively, as a result of water-calcite/aragonite fractionation (Grossman and Ku, 1986; Tremaine et al., 2011). This has the effect of slightly reducing the regional LGM $\delta^{18}O_{spel}$ signals, although the change is small and within the uncertainty of the regional signals. Enriched $\delta^{18}O_{precip}$ and $\delta^{18}O_{spel}$ values during the LGM must therefore be caused by a significant decrease in atmospheric moisture and

precipitation that resulted from the cooler conditions.

We have used version 2 of the SISAL database (Atsawawaranunt et al., 2018; Comas-Bru et al., 2020a) in our analyses. Despite the fact that SISALv2 includes more than 70% of known speleothem isotope records, there are still too few records from some regions (e.g. Africa, the Caribbean) to make meaningful analyses. The records for older time periods are also sparse. There are only 14 records from monsoon regions covering the LIG in SISALv2, for example. Nevertheless, our analyses show that there

are robust and explicable patterns for most monsoon regions during the Holocene and sufficient records to make meaningful analyses of the LGM and LIG. Whilst there is a need for the generation of new speleothem records from key regions such as northern Africa, further expansion of the SISAL database will certainly provide additional opportunities to analyse the evolution of the monsoons through time.

The impact of age uncertainties, included in SISALv2, are not taken into account in our analyses. Age uncertainties during the

Holocene are smaller than the interval used for binning records and the width of the time windows used, and thus should not have a significant effect on our conclusions. The mean age uncertainty at the LGM and LIG is ca 430 and 1140 years, respectively. However, varying the window length for the selection of LGM and LIG samples from ±500 to ±2000 years, thereby encompassing this uncertainty, has a negligible effect (<0.5 ‰) on the average $\delta^{18}O$ values. Thus, the interglacial-glacial contrast in regional $\delta^{18}O_{spel}$ is also robust to age uncertainties.

Isotope-enabled climate models are used in this study to explore observed regional-scale trends in $\delta^{18}O_{spel}$. There is a limited number of isotope-enabled models, and there are no simulations of the same time period using the same experimental protocol. Although there are simulations of the MH from both ECHAM5-wiso and GISS, for example, these models have different grid resolutions and used different boundary conditions. This could help to explain why the two models yield different estimates of the change in regional $\delta^{18}O_{precip}$ (of 0.5 ‰) at the MH. However, both models show trends in $\delta^{18}O_{precip}$ that reproduce the

observed changes in regional $\delta^{18}O_{spel}$ (Figs 3 and 4), and this provides a basis for using these models to explore the causes of these trends on different timescales. The failure to reproduce the LGM $\delta^{18}O_{spel}$ signal in SW-SAM in the ECHAM5-wiso model, which precluded a consideration of interglacial-glacial shifts in this region, is a common feature of other isotope-enabled simulations (Caley et al., 2014; Risi et al., 2010). Identifying the underlying relationships between $\delta^{18}O_{precip}$ and monsoon climate variables using multiple linear regression allows us to identify plausible mechanistic controls on $\delta^{18}O$

variability in the monsoon regions. Correlations between $\delta^{18}O$ and specific climate variables do not explicitly indicate causality. However, the relationships identified in the MLR model are consistent with the theoretical understanding of oxygen isotope systematics, and the findings of this paper are consistent with existing studies, suggesting that these relationships provide a plausible explanation for observed changes.

This study illustrates a novel data-model approach to investigate the relationship between $\delta^{18}O_{spel}$ and monsoon climate under

past conditions: We compare composite regional records and then use multiple linear regression of isotope-enabled palaeoclimate simulations to determine the change in individual climate variables associated with these trends. This obviates the need to use modern $\delta^{18}O_{precip}$-climate relationships to explain changes under conditions considerably different from today or to rely on coherency between different palaeohydrological archives which may respond to different climate variables. This model interrogation approach could be employed to address questions about the regional drivers of speleothem records outside

the monsoon regions.

**5 Conclusions**

Geographically distributed speleothem $\delta^{18}O$ records and isotope-enabled climate models can be used together to understand the underlying relationships between $\delta^{18}O_{spel}$ and monsoon climate in the past and therefore elucidate possible drivers of $\delta^{18}O$ variability. Speleothem records, objectively grouped into monsoon regions by record correlation and multivariate ordination

techniques, show regional trends that are consistent with changes in summer insolation but modulated by land-surface and ocean feedbacks. LGM $\delta^{18}O_{spel}$ signals are best explained by a large decrease in precipitation, as a consequence of lower atmospheric moisture content driven by global cooling. The evolution of $\delta^{18}O_{spel}$ through the Holocene across the global monsoon domain is closely correlated with changes in precipitation, atmospheric circulation and temperature. At the regional scale, our analyses support the increasing number of studies suggesting that East Asian monsoon speleothem $\delta^{18}O$ evolution

through the Holocene relates to changes in atmospheric circulation (i.e. changes in moisture pathway and/or source). Changes in regional precipitation are the predominant driver of Holocene $\delta^{18}O_{spel}$ evolution in the Indian, southwestern South American

and Indonesian-Australian monsoons, although changes in atmospheric circulation also contribute in the Indian and Indonesian-Australian monsoon regions and changes in precipitation recycling appear to be important in southwestern South America.


## Code and data availability

The SISAL (Speleothem Isotopes Synthesis and AnaLysis) database version 2 is available through the University of Reading Research Archive at http://dx.doi.org/10.17864/1947.256. The ECHAM5-wiso MH and LGM simulations are available at https://doi.org/10.1594/PANGAEA.902347. The ECHAM LIG simulation is available at

https://doi.pangaea.de/10.1594/PANGAEA.879229. The OIPC mean annual $\delta^{18}O_{precip}$ data are available at http://wateriso.utah.edu/waterisotopes/pages/data_access/ArcGrids.html. CRU-TS4.01 mean annual temperature data are available at http://doi.org/10/gcmcz3. The GISS simulations and code used to generate the figures in this paper are available at https://doi.org/10.5281/zenodo.3875496.

## Author contributions

The study was designed by SP, SPH, LCB and NK. MW and ALG provided climate model outputs. SP ran the analyses. The first draft of the manuscript was written by SP, SPH and LCB and all authors contributed to the final draft.

## Competing interests

The authors declare no competing interests.

## Funding

SP, LCB and SPH acknowledge funding support from the ERC-funded project GC2.0 (Global Change 2.0: Unlocking the past for a clearer future, grant number 694481). SPH also acknowledges support from the JPI-Belmont Forum project "Palaeoclimate Constraints on Monsoon Evolution and Dynamics (PaCMEDy)" funded through NERC (NE/P006752/1).

## Acknowledgements

Ideas in this paper were developed at a meeting of the SISAL (Speleothem Isotopes Synthesis and Analysis) working group

of the Past Global Changes (PAGES) programme. We thank PAGES for their support for this meeting and our colleagues in

SISAL for the useful discussions. We thank Gabriele Messori for help with the code for calculating precipitation recycling. We also thank Steven Clemens and two anonymous reviewers for their helpful comments on this paper.

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

925

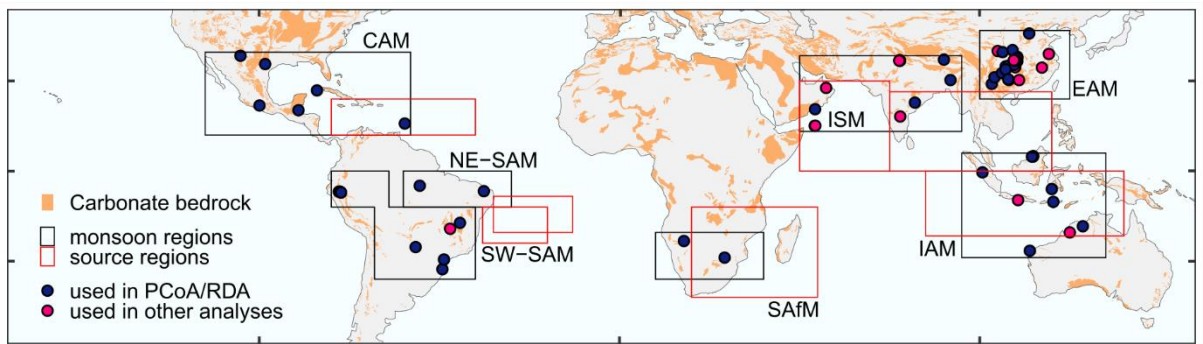

**Figure 1: Spatial distribution of speleothem records used is this study. Colours indicate the sites used in Principal Coordinates Analysis and Redundancy Analysis (PCoA, RDA) to separate monsoon regions, and sites not used in PCoA and RDA but used in subsequent analyses. The individual regional monsoons are shown by boxes: CAM = Central American Monsoon (latitude: 10 to 33°; longitude: -115° to -58°), SW-SAM = southwestern South American Monsoon (latitude: -10° to 0°; longitude: -80° to -64° and latitude: -30° to -10°; longitude -68° to -40°), NE-SAM = northeastern South American Monsoon (latitude: -10° to 0°; longitude: -60° to -30°), SAfM = southern African Monsoon (latitude: -30° to -17°; longitude: 10° to 40°), ISM = Indian Summer Monsoon (latitude: 11° to 32°; longitude: 50° to 95°), EAM = East Asian Monsoon (latitude: 20° to 39°; longitude: 100° to 125°), IAM = Indonesian-Australian Monsoon (latitude: -24° to 5°; longitude: 95° to 135°). Source region limits used in the multiple linear regression analysis are also shown. The background carbonate lithology is from the World Karst Aquifer Mapping (WOKAM) project (Goldschneider et al., 2020).**

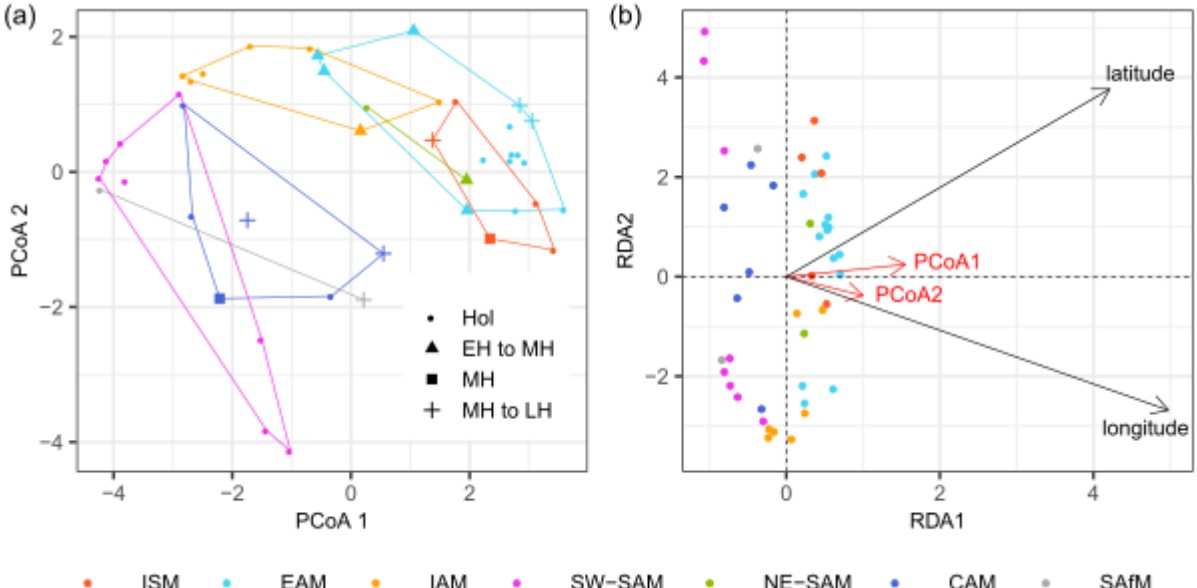

**Figure 2: Results of Principal Coordinate Analysis (PCoA) and Redundancy analysis (RDA). (a) PCoA biplot showing the loadings of each site on the first 2 axes, which represent 85% of the total variance. Shapes indicate the Holocene coverage of each site, where sites with a coverage ≥ 8000 years represent most or all of the Holocene (Hol). Sites with a temporal coverage of < 8000 years are coded to show whether they represent the early to mid-Holocene (EH to MH, record midpoint > 8,000 years BP), the mid Holocene (MH, record midpoint between 8,000 and 5,000 years BP), or the mid to late-Holocene (LH to MH, midpoint <5,000 years BP). (b)**

RDA triplot, where the response variables are the PCoa1 and PCoA2 axes explained by latitude and longitude. The direction of the PCoA axes have been fixed so that they align with the explanatory variables.

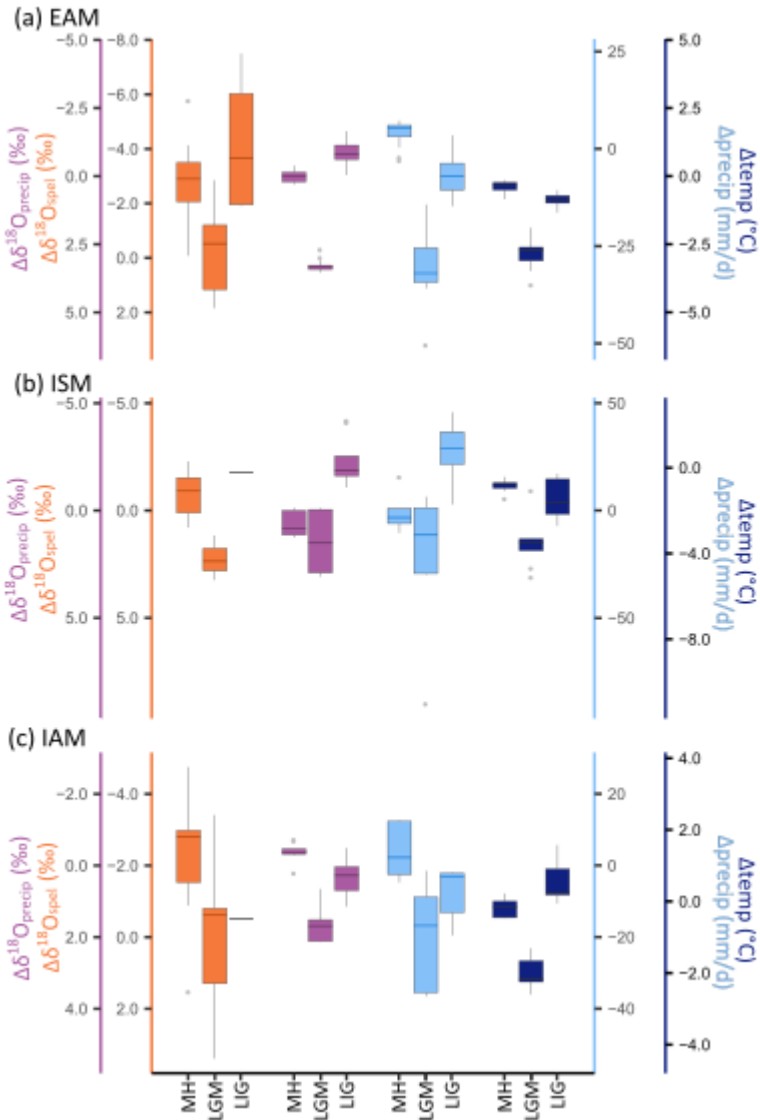

 Figure 3: Speleothem δ18O anomalies compared to anomalies of δ18Oprecip, precipitation and temperature from the ECHAM simulations for the (a) East Asian (EAM), (b) Indian (ISM) and (c) Indonesian-Australian (IAM) monsoons. The boxes show the median value (line) and the interquartile range, and the whiskers show the minimum and maximum values, with outliers represented by grey dots. Note that the isotope axes are reversed, so that the most negative anomalies are at the top of the plot, to be consistent with the assumed relationship with the direction of change in precipitation and temperature.

955

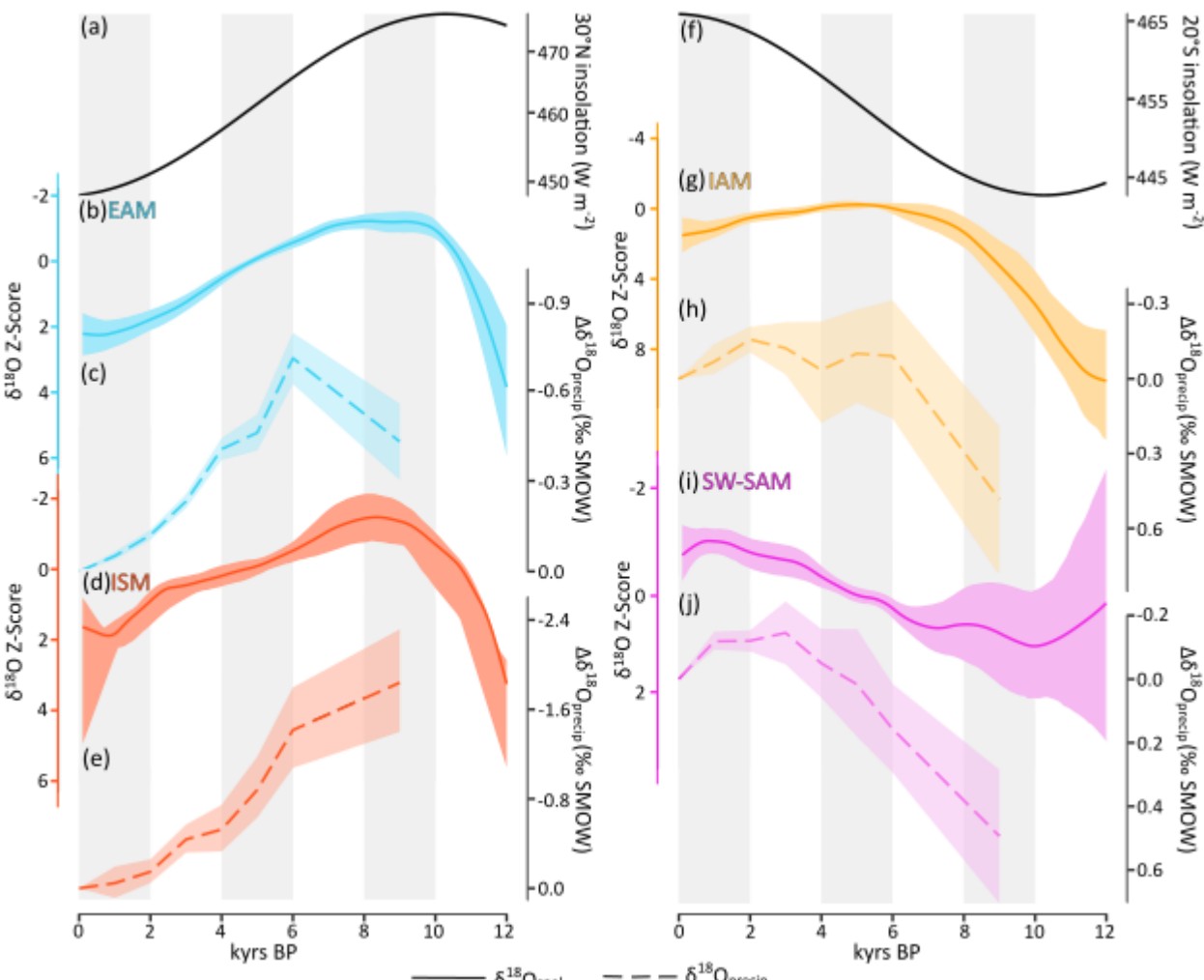

**Figure 4: Evolution of regional speleothem δ18Oprecip signals through the Holocene compared to δ18Oprecip simulated by the GISS model.** The left panel shows northern hemisphere monsoons (EAM = East Asian Monsoon; ISM = Indian Summer Monsoon) and summer (May through September) insolation at 30° N (Berger, 1978). The right panel shows southern hemisphere monsoons (SW-SAM = southwest South American Monsoon; IAM = Indonesian-Australian Monsoon) and summer (November through March) insolation for 20° S (Berger, 1978). The speleothem δ18O changes are expressed as z-scores, with a smoothed loess fit (3,000 year window), and confidence intervals obtained by bootstrapping by site. δ18Oprecip values are expressed as anomalies from the pre-industrial control simulation. Note that the isotope axes are reversed, so that the most negative anomalies are at the top of the plot, to be consistent with the assumed relationship with the changes in insolation.

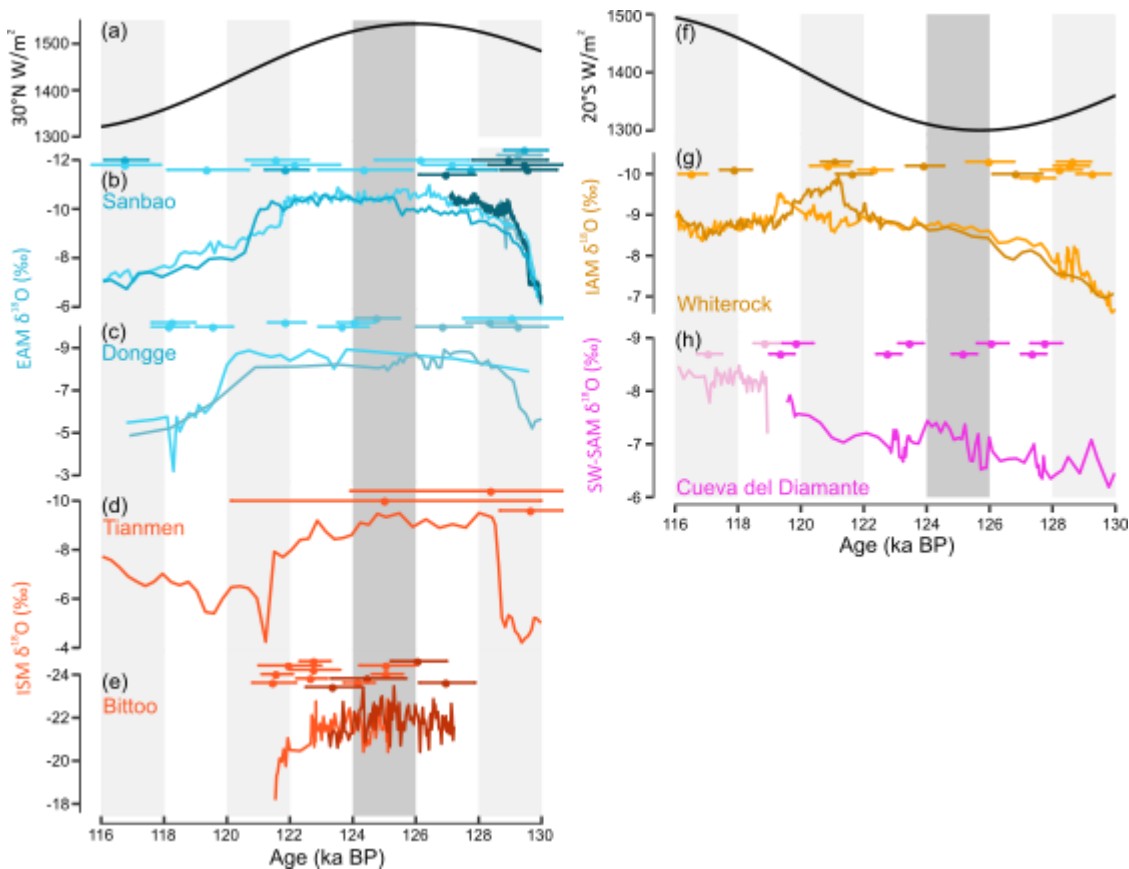

**Figure 5: Comparison of changes in summer insolation and δ¹⁸O$_{spel}$ through the peak of the Last Interglacial (Stage 5e) from the (b,c) East Asian Monsoon (EAM), (d,e) Indian Summer Monsoon (ISM), (g) southwest South American Monsoon (SW-SAM) and (h) Indonesian-Australian Monsoon (IAM) regions. The U/Th dates and uncertainties are shown for each record. The summer insolation curves (Berger, 1978) are for May through September at 30° N in the northern hemisphere (a) and for November through March for 20° S in the southern hemisphere (f). Note that the isotope axes are reversed, so that the most negative anomalies are at the top of the plot, to be consistent with the assumed relationship with the changes in insolation. The LIG (Stage 5e) time slice used in the analysis in section 2.4 is shown by the dark grey bar.**

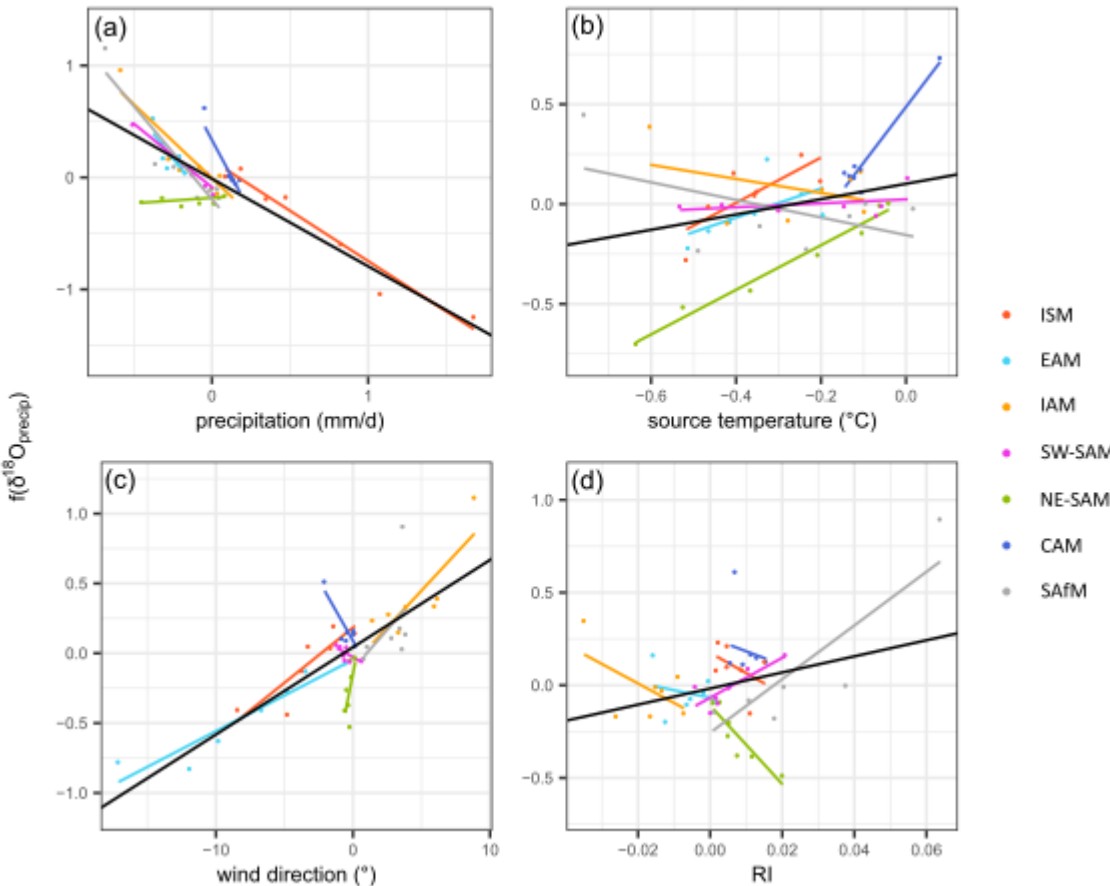

**Figure 6: Partial residual plots from the multiple linear regression analysis, showing the relationship between anomalies in simulated δ¹⁸O$_{precip}$ and the four predictor variables, after taking account of the fitted partial effects of all the other predictors. The simulated δ¹⁸O$_{precip}$ are anomalies relative to the pre-industrial control simulation, and are annual values weighted by precipitation amount. The predictor variables are: precipitation in the delineated monsoon region (mm/d), temperature in the source region (°C), surface wind direction over the source region (°) as an index of potential changes in source region and the ratio of precipitation recycling to total precipitation over the monsoon region (RI, unitless). The predictor variables are summer mean values, representing the summer monsoon, where summer is defined as May to September for northern hemisphere monsoons and November to March for southern hemisphere monsoons.**

| | PCoA1 | PCoA2 | PCoA3 | PCoA4 | PCoA5 |
|---|---|---|---|---|---|
| Eigenvalue | **269.06** | **85.22** | 16.81 | 10.25 | 5.55 |
| Explained (%) | **64.87** | **20.55** | 4.054 | 2.47 | 1.34 |
| Cumulative (%) | **64.87** | **85.42** | 89.48 | 91.95 | 93.27 |

**Table 1: Results of the Principal Coordinates Analysis (PCoA). Significant axes, as determined by the broken stick method (Bennett, 1996), are shown in bold.**

|            | RDA1     | RDA2  |
|------------|----------|-------|
| Latitude   | **0.88** | -0.47 |
| Longitude  | **0.75** | 0.67  |
| Eigenvalue | 0.73     | 0.04  |
| Explained (%) | 36.7  | 2.2   |

**Table 2: Results of the redundancy analysis (RDA). Variables that are significantly correlated (P < 0.01) with the RDA axes are shown in bold.**

|                          | Regression coefficient | T value |
|--------------------------|------------------------|---------|
| Regional precipitation   | **-0.78**              | -8.75   |
| Source area temperature  | **0.39**               | 2.05    |
| Wind direction           | **0.06**               | 8.03    |
| Precipitation recycling  | 4.34                   | 1.92    |

**Table 3: Results of the multiple linear regression analysis. Significant relationships (P < 0.01) are shown in bold.**

995

1000