# Peer review of "A data-model approach to interpreting speleothem oxygen isotope records from monsoon regions"

_Climate of the Past, 2020_

## Referee Comment (RC1) · Anonymous Referee #1 · 27 Jul 2020

The article investigates the water isotopic variations recorded in speleothems in monsoon regions worldwide and on orbital time scales. It uses the SISAL database (Atsawawaranunt et al. (2018)) and global simulations from two general circulation models (ECHAM and GISS).

I think this paper is an interesting contribution to this active area of research. However, my feeling at the end of the paper is that it is an addition of statistical diagnostics, and I'm often lacking an overall view and physical interpretations. In addition, when the model simulation are used, I wish I had some evaluation and discussion on to what extent the models can be trusted.

[Figure]

**1   Major comments**

**1.1   Give some overall view and more physical intepretation**

The paper directly dives into complicated statistical diagnostics. But before this, I think some overview would be useful. For example, a few basic figures showing the maps of simulated $\delta^{18}O_{precip}$ anomalies for a few key periods would be useful before showing the regional averages.

When describing the results of the statistical methods, it would be useful to better guide the reader in the physical interpretation of the figures: what does it mean when values are more negative, positive, larger, smaller... (more details in minor comments.)

At the end of each sub-section, a few sentences would be useful to summarize the results in terms of physical understanding of the processes driving the isotopic variability. A statistical analysis is not enough to identify causality and thus isotopic "drivers", so the discussion should rely more on the huge body of litterature devoted to the interpretation of water isotopic records in monsoon regions.

**1.2   Evaluate and discuss the model realism and robustness**

The models are used in the regression analysis but what is the realism of the simulations? To what extent can they be trusted?

Some comparison between SISAL and the models are shown in the figures, but the variables and diagnostics are different, so it's hard to compare (more details in minor comments). The observations and simulations should be compared in a more rigourous way. Also, figure 1 could be redone with the models, as an additional check of the realism of the simulations. An entire sub-section should be devoted to model evaluation.
Every time it is possible, both models should be used for the same diagnostics to assess te robustness. It's a great opportunity to have two models, and it should be used more systematically.

After the evaluation section, the reader should have a clear opinion on what feature in the simulations can or cannot be trusted. Then when the regression analysis is performed, there should be some discussion on what specific results can be trusted or not.

**2 Minor comments**

- l 48: "The temperature effects stem from the temperature dependance of oxygen isotope fractionation during condensation and ..." -> "The temperature effects stem from the oxygen isotope fractionation during condensation and ...". The contribution of the temperature dependance of the fractionation coefficient in the temperature effect is small (e.g. realistic results can be obtained even with constant isotopic fractionation: Galewsky and Hurley (2010)).

- l 59: "depleted" -> "enriched"? Actually, it depends depleted or enriched compared to what, but the specificity of evapo-transpiration is to be enriched relatively to the overlying water vapor, and thus to have an enriching effect of the water vapor (Gat and Matsui (1991)).

- l 64: you can also add Caley et al. (2014) in the citations.

- l 189: define "OIPC": is it the dataset described above?

- Figure 3: I have trouble reading this figure. For $\delta^{18}O$, is it possible to have the same y-scale for $\Delta\delta^{18}O_{precip}$ and $\Delta\delta^{18}O_{spel}$ ? This would allow a direct visual comparison of these 2 quantities. I also have trouble seeing whether anomalies

are negative or positive: could you draw an horizontal line to indicate the 0? The 0 line could be shared for all potted variables.

In addition, why do you compare observed $\Delta\delta^{18}O_{spel}$ to simulated $\Delta\delta^{18}O_{precip}$? Why not converting simulated $\Delta\delta^{18}O_{precip}$ into $\delta^{18}O_{calcite}$ for a more rigorous comparison?

- l 270: "consistently low PCoA1 scores": what does it physically mean?

- l 300: "The regional composites are z-scores, i.e. anomalies with respect to the base period (3000-7000 yr BP)." Are these just anomalies or true z-scores? Please clarify how you calculate those z-scores and what they physically mean. And why using z-scores in the first place? Why not just simple anomalies?

- Fig 4: what are the units of the plotted variables? Please add the units on the y-labels. I have trouble to compare the simulated and observed $\delta^{18}O$: please use similar diagnostics and units for both. For example, convert precip $\delta^{18}O$ into calcite $\delta^{18}O$ for the model, and use simple $\delta^{18}O$ anomalies for the speleothem observations.

- Fig 6: can you explain better how these diagrams should be interpreted? What do they physically mean?

- Fig 6, section 3.4: on which model was this regression analysis done? GISS or ECHAM? More generally, why doing each diagnostic with only one model? Why not doing each diagnostic with each model (when the period of interest is available), to assess the robusntess of the results?

- l 380: "drivers" -> "meteorological variables". This is just a statistical analysis, so no causality can be identified, so the meteorological variables cannot be assumed to be drivers.

- l 389: "changes in precipitation amount" -> "changes in local precipitation amount": changes in upstream precipitation amount has been shown to be very important in previous studies (e.g. Battisti et al. (2014)) but were not analyzed here.

- Table 1: too many digits in the numbers.

**References**

Atsawawaranunt, K., Comas-Bru, L., Amirnezhad Mozhdehi, S., Deininger, M., Harrison, S. P., Baker, A., Boyd, M., Kaushal, N., Ahmad, S. M., Ait Brahim, Y., et al. (2018). The sisal database: A global resource to document oxygen and carbon isotope records from speleothems. *Earth System Science Data*, 10(3):1687–1713.

Battisti, D., Ding, Q., and Roe, G. (2014). Coherent pan-asian climatic and isotopic response to orbital forcing of tropical insolation. *Journal of Geophysical Research: Atmospheres*, 119(21):11–997.

Caley, T., Roche, D. M., and Renssen, H. (2014). Orbital asian summer monsoon dynamics revealed using an isotope-enabled global climate model. *Nature communications*, 5(1):1–6.

Galewsky, J. and Hurley, J. V. (2010). An advection-condensation model for subtropical water vapor isotopic ratios. *J. Geophys. Res.*, 115 (D16):D16115 , doi:10.1029/2009JD013651.

Gat, J. R. and Matsui, E. (1991). Atmospheric water balance in the Amazon basin: An isotopic evapotranspiration model. *J. Geophys. Res.*, 96:13179–13188.

---

## Referee Comment (RC2) · Anonymous Referee #2 · 4 Aug 2020

This manuscript studies the speleothem oxygen isotope ($\delta$18O) records in monsoon regions worldwide on orbital timescales by using the SISAL database along with the isotope-enabled climate model simulations. It is indeed an important approach which may gain new insights into speleothem $\delta$18O interpretation via data-model comparison/statistical analyses. However, the current manuscript has several issues to clarify and/or improve before it can be considered for publishing in Climate in the Past.

Major comments:

(1) The introduction does not reflect the current understanding of the speleothem $\delta$18O, particularly in the East Asian monsoon domain. For example, it basically follows the

previous misunderstanding(s) from modeling and other research communities, especially on orbital-scale, that the speleothem $\delta18O$ was interpreted as a rainfall amount proxy by the Chinese speleothem community over the past two decades. In fact, the mainstream idea form the speleothem community has never been the 'amount affect' (e.g., Cheng et al., 2019), and therefore, one of main scientific issues addressed here is groundless.

(2) The authors mentioned that "a composite record can minimize the influence of site-specific karst and cave processes" (with real spatial variations?). However, the results and/or assumptions from the PCoA method are tentative, which lacks a underlying mechanism. The same monsoon system (e.g., the ISM and EAM boxes in the figure 1) could have different speleothem $\delta18O$ patterns on orbital-scale, as illustrated by a number of modeling results (e.g., Liu et al., 2014; Battisti et al., 2014).

(3) Lines 66-68: This is really a misleading statement. I suggest that the authors should read the original papers they cited here more carefully (as well as Cheng et al., 2016, 2019; Zhang et al., 2018; Zhang et al., 2020) and quote the original statements in these papers if necessary. For example, Cheng et al. (2009) (cited in the sentence) clearly asserted: "Thus, neither the temperature- $\delta18O$ relationship, commonly used to interpret ice-core data, nor the interpretation based on the "amount effect" is justified".

(4) Lines 229-236 and figure 4: What are the simulated precipitation $\delta18O$ values in the EAM, ISM, IAM, SW-SAM domains? Are they amount-weighted annual mean precipitation $\delta18O$ values, annual mean precipitation $\delta18O$ values or only summer (MJJAS) mean precipitation $\delta18O$ values? In addition, please give the boundary coordinate of these monsoon regions (the EAM, ISM, IAM, SW-SAM...) for the calculations. Give a detail explanation about the $\delta18O$ amplitude differences between observation and model results in the figure 4 if significant.

(5) Lines 376-379: "...there is little different in the $\delta18O$ values between the MH and the LIG in the ISM and EAM regions...", "Given that the increase in summer insolation

is much larger during the LIG than the MH, this finding is again consistent with the idea that other factors play a role in modulating the monsoon response to insolation forcing". What are the other factors and the processes? Moisture source and/or pathway? Or some kind of thresholds (e.g., Cheng et al., 2012; Cai et al., 2015)? In addition, the summer insolation is indeed higher during the LIG than during the MH, but the monsoon circulation or intensity is influenced by the temperature (thus pressure?) gradient between land and sea as well. What is the difference of the land-sea temperature (pressure) gradients for the MH and the LIG periods? Or monsoon circulation scales? A more comprehensive discussion of the issue with a help of climate models would be very welcome.

(6) The main conclusion is that "East Asian monsoon speleothem $\delta18O$ evolution through the Holocene relates to changes in atmospheric circulation (i.e. changes in moisture pathway and/or source). Changes in precipitation amount are the predominant driver of Holocene $\delta18Ospel$ evolution in the Indian, southwestern South American and Indonesian-Australian monsoons, although changes in atmospheric circulation also contribute in the Indian and Indonesian-Australian monsoon regions and changes in precipitation recycling in southwestern South America". This conclusion is not well supported and problematic as well. First, the 'amount effect' discussed here is not the same 'amount effect' as conventionally defined in the tropics (see Zhao et al., 2019 for instance). The authors implies that the local rainfall amount drive the orbital-scale variations in speleothem $\delta18O$ value. They really need to provide a mechanism/calculation for the Indian, southwestern South American and Indonesian-Australian monsoon systems to explain how the oxygen isotopic fractionation under different conditions of rainfall amounts at each cave site could result in the observed $\delta18Ospel$ changes on orbital-scale without significant monsoon circulation (including the moisture pathway and/or source) changes. On the other hand, the "East Asian monsoon speleothem $\delta18O$ evolution through the Holocene relates to changes in atmospheric circulation" is just a reinforcement of the previous view on the East Asian monsoon evolution inferred by speleothem $\delta18O$ records published in a large number of speleothem works over

the past two decades. In short, it is the monsoon circulation that to first order drives the orbital $\delta$18Ospel changes, not only for the East Asian monsoon, but also (most likely) for other monsoon systems.

(7) Please illustrate the x- and y-axes of the figure 2a in the section 3.1 or describe them in the section 2.3. In the section 3.1, the authors illustrated that Southern Hemisphere monsoon regions are characterized by low PCoA1 scores, while Northern Hemisphere monsoon regions are characterized by higher PCoA1 scores. Please explain these terms in the context of instrumental data or modern climatology, which may be more interesting for the paleoclimate community.

(8) The authors used the anomaly for comparison from different model results. However, readers might also want to see a detailed comparison between model results, particularly between the model results from this study and those from previous studies.

(9) Lines 397-400 and the figure 3: "The LGM is characterised by a similar orbital configuration to today, however global ice volume was at a maximum and GHG concentrations were lower than present. The $\delta$18Ospel anomalies are more positive during the LGM than the MH or LIG, suggesting drier conditions in the ISM, EAM and IAM, supported by simulated changes in $\delta$18Ospel and precipitation (Fig. 3)." This sentence is again misleading. While the authors highlighted a similar orbital configuration between the LGM and today, they actually discussed the issue related to a comparison of the LGM with the MH or LIG, presumably implying that they have similar orbital configurations. The LGM (21±1ka) is near a Northern Hemisphere insolation minimum whereas the MH/LIG are near the insolation maxima. As such the related discussions should be rephrased, and so does the related conclusion, since the insolation difference should be taken into account together with GHG and the global ice volume, because one could also argue that the $\delta$18Ospel just follows the insolation with effect to a lesser extent from GHG and the global ice volume.

Minor comments:

Lines 97, 112 and 160, 'the Principal Coordinate Analysis (PCoA)', the abbreviation occurred three times, keep the first one.

Line 121, please give the full name of the climate models: ECHAM5 and GISS E-R

Line 163, '...missing data that ...' , 'that' should be 'than'?

Line 189, what is the 'OIPC'?

Lines 268-277, the abbreviations (EAM, SW-SAM, SAfM, CAM, IAM) occurred too late in the section 3.1, it's better put them in the introduction.

Line 358 'southern China Sea' should be 'South China Sea'.

Figure 5, the time series for Dongge Cave can be replaced by a high-resolution time-series, please double check with the database.

References:

Cheng, H., Zhang, H., Zhao, J., Li, H., Ning, Y., & Kathayat, G., 2019. Chinese stalagmite paleoclimate researches: A review and perspective. Science China Earth Sciences, 62(10), 1489-1513. doi: 10.1007/s11430-019-9478-3.

Zhao, J.Y., Cheng, H., Yang, Y., Tan, L.C., Spötl, C., Ning, Y.F., Zhang, H.W., Cheng, X., Sun, Z., Li, X.L., Li, H.Y., Liu, W., Edwards, R.L., 2019. Reconstructing the western boundary variability of the Western Pacific Subtropical High over the past 200 years via Chinese cave oxygen isotope records. Clim. Dyn 52, 3741–3757.

Cheng, H., Edwards, R.L., Wang, Y., Kong, X., Ming, Y., Kelly, M.J., Wang, X., Gallup, C.D., 2006. A penultimate glacial monsoon record from Hulu Cave and two-phase glacial terminations. Geology 3 (34), 217–220.

Cheng, H., Edwards, R.L., Sinha, A., Spötl, C., Yi, L., Chen, S., Kelly, M., Kathayat, G., Wang, X., Li, X., Kong, X., Wang, Y., Ning, Y., Zhang, H., 2016a. The Asian monsoon over the past 640,000 years and ice age terminations. Nature 534 (7609), 640–646.

Cheng, H., Sinha, A., Wang, X., Cruz, F.W., Edwards, R.L., 2012. The global paleo-monsoon as seen through speleothem records from Asia and the Americas. Clim. Dynamis 39 (5), 1045–1062.

Cai, Y., Fung, I. Y., Edwards, R. L., An, Z., Cheng, H., Lee, J. E., Tan, L., Shen, C. C., Wang, X., Day, J. A., Zhou, W., Kelly, M. J. and Chiang, J. C. H., 2015. Variability of stalagmite-inferred Indian monsoon precipitation over the past 252,000 y, Proc. Natl. Acad. Sci. U. S. A., 112(10), 2954–2959.

Battisti, D. S., Ding, Q. and Roe, G. H., 2014. Coherent pan-Asian climatic and isotopic response to orbital forcing of tropical insolation, J. Geophys. Res. Atmos., 119(21), 11997-12020, https://doi.org/ 10.1002/2014JD021960.

Liu, Z., Wen, X., Brady, E.C., Otto-Bliesner, B., Yu, G., Lu, H., Cheng, H., Wang, Y., Zhang, W., Ding, Y., Edwards, R.L., Cheng, J., Liu, W., Yang, H., 2014. Chinese cave records and the east asia summer monsoon. Quat. Sci. Rev. 83, 115–128.

Cheng, H., Edwards, R.L., Broecker, W.S., Denton, G.H., Kong, X., Wang, Y., Zhang, R., Wang, X., 2009. Ice age terminations. Science 326 (5950), 248–252.

Zhang, H., Griffiths, M.L., Chiang, J., Kong, W., Wu, S., Atwood, A., Huang, J., Cheng, H., 2018. East Asian hydroclimate modulated by the position of the westerlies during Termination I. Science 362, 580–583.

Zhang, H., Cheng, H., Baker, J., & Kathayat, G., 2020. Response to Comments by Daniel Gebregiorgis et al. "A Brief Commentary on the Interpretation of Chinese Speleothem $\delta$18O Records as Summer Monsoon Intensity Tracers". Quaternary 2020, 3, 7. Quaternary, 3(1), 8.

---

## Author Comment (AC1) · 9 Oct 2020

Response to referee #1

**Major comments**

1.1 Give some overall view and more physical interpretation

The paper directly dives into complicated statistical diagnostics. But before this, I think some overview would be useful. For example, a few **basic figures showing the maps** of simulated δ 18Oprecip anomalies for a few key periods would be useful before showing the regional averages. When describing the results of the statistical methods, it would be useful to better guide the reader in the **physical interpretation of the figures**: what does it mean when values are more negative, positive, larger, smaller... (more details in minor comments.) At the end of each sub-section, a few sentences would be useful to **summarize the results** in terms of physical understanding of the processes driving the isotopic variability. A statistical analysis is not enough to identify causality and thus isotopic "drivers", so the discussion should rely more on the huge body of literature devoted to the interpretation of water isotopic records in monsoon regions.

*The focus of this paper is not to use isotope-enable models directly to explain observed changes in the speleothem records but rather to use statistical approaches to explore patterns in the observations and model outputs, on the assumption that consistency between the two reveals physically plausible explanations of regional speleothem changes. We will modify the introduction to make the logic of using statistical analyses combining observations and model outputs clearer (please see specific modifications below). Since we are using previously published model results, it does not seem necessary to include a separate section describing these results. However, we will include anomaly maps of simulated $\delta^{18}O_{precip}$ and speleothem $\delta^{18}O$ data from SISAL v2 in the supplementary material and refer to these maps in the main text.*

*Although the statistical approaches we are using (PCoA, multiple regression, z-scores) are not commonly applied to speleothem records, they are standard techniques for analysing other kinds of environmental data. However, to guide readers through these statistical analyses, we will revise the text in the methods and results sections to clearly describe what each analysis or figure shows, as follows:*

*Section 2.3 (line 167): "PCoA results were displayed as a biplot, where sites ordinated close to one another (i.e., with similar PCoA scores) show similar trends and sites ordinated far apart have dissimilar trends."*

*Section 3.1 (line 268): "PCoA shows the (dis)similarity of Holocene $\delta^{18}O_{spel}$ evolution across individual records, and thus allows an objective regionalisation of these records."*

*Section 3.2 (line 280): "To investigate the causes of glacial-interglacial shifts in $\delta^{18}O$, we compare simulated and observed regional $\delta^{18}O$ signals during the LIG, LGM and MH with shifts in climate variables (precipitation and temperature)."*

*Section 3.4 (line 319): "The MLR analyses of simulated $\delta^{18}O_{precip}$ trends identify the impact of an individual climate variable on $\delta^{18}O_{precip}$ in the absence of changes in other variables."*

*We agree with the reviewer that statistical relationships do not necessarily indicate causal relationships. Generally, explanations of the causes of observed $\delta^{18}O_{spel}$ variability either rely on modern $\delta^{18}O$-climate observations and assume that these are constant through time (e.g. Sinha et*

*al., 2015), or interpret changes by comparison with other palaeoclimate reconstructions (e.g. Maher, 2008; Ward et al., 2019). However, $\delta^{18}$O-climate relationships may not have remained constant through the past and cross-comparison between palaeoclimate reconstructions is complicated by the fact that different archives record climate in different and non-linear ways.  We therefore tackle this problem with a data-model approach that has two main advantages:*

- *By using a large number of coexistent speleothem records to identify the large-scale coherent trends, we reduce the impact of non-climatic factors (i.e. soil and karst processes) on $\delta^{18}O_{spel}$. This approach focuses on the trends consistent across records that are inherited from $\delta^{18}O_{precip}$.*
- *We use model simulations that explicitly include water isotope physics to reproduce large-scale orbital trends in $\delta^{18}$O. These therefore provide a physically plausible explanation of $\delta^{18}$O trends under past climate conditions. Congruence between the observed and simulated trends suggests that the drivers of regional changes in the model world are plausible explanations of these changes in the observations. Multiple regression analysis is a convenient way of exploring the various drivers of $\delta^{18}$O.*

*Our approach to explain past isotope changes in terms of specific climate drivers is robust as it takes into account large-scale trends in $\delta^{18}O_{spel}$ using a known understanding of isotope physics.*

*To clarify how our approach investigates the underlying mechanisms of $\delta^{18}O_{spel}$ trends, we will amend the introduction (from line 79):*

"These interpretations generally rely on modern $\delta^{18}O_{precip}$-climate observations, which may not have remained constant through time. The sources of $\delta^{18}$O variability can also be explored using isotope-enabled climate models (e.g. Hu et al., 2019), which incorporate known isotope effects and therefore provide plausible explanations for $\delta^{18}O_{spel}$ trends."

*And from line 91:*

"In this study, we combine speleothem $\delta^{18}$O records from version 2 of the Speleothem Isotopes Synthesis and Analysis (SISAL) database with isotope-enabled palaeoclimate simulations from two climate models to investigate the plausible mechanisms driving changes in $\delta^{18}$O in monsoon regions through the Holocene (last 11,700 years) and between interglacial (mid-Holocene and Last Interglacial) and glacial (Last Glacial Maximum) states."

*Given the inherent limitations, discussed above, in interpreting the speleothem records based on modern relationships and/or comparison with other reconstructions, our statistical approach offers new insights into the interpretation of regional changes. Nevertheless, we have included a discussion (from line 380 to 396) of how our results fit with existing literature.*

1.2  Evaluate and discuss the model realism and robustness

The models are used in the regression analysis but what is the realism of the simulations? To what extent can they be trusted? Some comparison between SISAL and the models are shown in the figures, but the variables and diagnostics are different, so it's hard to compare (more details in minor comments). The observations and simulations should be compared in a more rigorous way. Also, figure 1 could be redone with the models, as an additional check of the realism of the simulations. An entire sub-section should be devoted to model evaluation. Every time it is possible, both models should be used for the same diagnostics to assess to robustness. It's a great opportunity to have two

models, and it should be used more systematically. After the evaluation section, the reader should have a clear opinion on what feature in the simulations can or cannot be trusted. Then when the regression analysis is performed, there should be some discussion on what specific results can be trusted or not.

*Climate models produce internally physically consistent changes in the simulated variables and our goal in this paper is not to evaluate the model simulations as such. This has been done to a greater or lesser extent in previous publications (e.g. LeGrande and Schmidt, 2009; Wackerbarth et al., 2012, Gierz et al., 2017; Werner et al., 2018; Comas-Bru et al., 2019). Here we assume that the broadscale trends shown in these simulations are robust and that they can be used to diagnose what factors might contribute to observed changes in speleothem δ^18O between glacial and interglacial states and through the Holocene. We will make this clearer by rewriting the sentence in line 101 to explain this logic as follows:*

"We exploit the fact that models produce internally physically consistent changes to explore potential and plausible causes of the trends observed in speleothem records across specific monsoon regions, using multiple regression analysis."

*Since we are using previously published simulations, our description of the models focuses on the model set-up boundary conditions. However, we agree that it would be worthwhile to expand these descriptions in order to comment on their reliability on the basis of previously published analyses and will amend the model description text. We will also include figures showing relevant model outputs in Supplementary.*

*We agree that directly comparing multiple models would be a good way to test the robustness of our findings, but this is currently not possible. There are only a few isotope-enabled palaeoclimate simulations and they often use different modelling protocols. Here, for example, the only time period which was run by both models was the mid-Holocene (6 ka) and the experimental protocols by each modelling group were different. This makes it difficult to isolate the reasons behind any differences between the two simulations (~0.5‰, line 425). This is why we decided to focus on comparing glacial-interglacial trends using the ECHAM simulations, and the trends through the Holocene using the GISS simulations. We will re-order and rewrite the model description section (section 2.2; from line 121) to make this logic clearer as follows:*

[revised manuscript text omitted]

**Minor comments**

- l 48: "The temperature effects stem from the temperature dependance of oxygen isotope fractionation during condensation and ..." -> "The temperature effects stem from the oxygen isotope fractionation during condensation and ...". The contribution of the temperature dependance of the fractionation coefficient in the temperature effect is small (e.g. realistic results can be obtained even with constant isotopic fractionation: Galewsky and Hurley (2010)).

*We will reword this sentence as follows: "The temperature effect stems from the cooling required for progressive rainout during Rayleigh distillation (Dansgaard, 1964; Rozanski et al., 1993)."*

- l 59: "depleted" -> "enriched"? Actually, it depends depleted or enriched compared to what, but the specificity of evapo-transpiration is to be enriched relatively to the overlying water vapor, and thus to have an enriching effect of the water vapor (Gat and Matsui (1991)).

*We will reword this sentence as follows: "The isotopic composition of atmospheric water vapour may also be modified by precipitation recycling over land, since evapotranspiration returns moisture from precipitation back to the atmosphere thereby reducing the $\delta^{18}O_{precip}$/distance gradient along an advection path that occurs with Rayleigh distillation (Gat, 1996; Salati et al., 1979)."*

- l 64: you can also add Caley et al. (2014) in the citations.

*This section of the introduction summarises the various ways speleothem $\delta^{18}O$ records were interpreted in the original publications. Caley et al. (2014) does not publish or interpret a new speleothem d18O record but is a model-based analysis of factors driving changes in the Asian monsoon $\delta^{18}O_{spel}$ using an isotope-enabled model. We do not think it is relevant to cite it here.*

- l 189: define "OIPC": is it the dataset described above?

*Yes, this is the data set described in line 185-186. We apologise for not naming it there and will amend the text to do so, as follows: "... using as reference the Online Isotopes in Precipitation Calculator (OIPC: Bowen, 2018; Bowen and Revenaugh, 2003), a global gridded dataset of interpolated mean annual precipitation-weighted $\delta^{18}O_{precip}$ data."*

- Figure 3: I have trouble reading this figure. For δ18O, is it possible to have the same y-scale for Δδ 18Oprecip and Δδ 18Ospel? This would allow a direct visual comparison of these 2 quantities. I also have trouble seeing whether anomalies are negative or positive: could you draw an horizontal line to indicate the 0? The 0 line could be shared for all potted variables. In addition, why do you compare observed Δδ 18Ospel to simulated Δδ 18Oprecip? Why not converting simulated Δδ 18Oprecip into δ 18Ocalcite for a more rigorous comparison?

*In figure 3, each variable has different units and axes have been adjusted so that glacial-interglacial patterns are aligned for easier reading of trends, rather than comparison of quantitative values. Adding zero lines for each variable would make the figure more difficult to read. However, we will modify it so that boxes are grouped together by variable ($\delta^{18}O_{spel}$, $\delta^{18}O_{precip}$, precipitation, temperature) instead of by time period. We will order each group by time slice (MH, LGM, LIG). We think that this will make it easier for readers to see and interpret the trends. The $\delta^{18}O_{precip}$ has not been converted to its speleothem-equivalent (i.e., $\delta^{18}O_{calcite}$) as this requires knowing mean cave temperature which would have to be estimated by using model-simulated temperature, thereby adding more uncertainty to the data.*

- l 270: "consistently low PCoA1 scores": what does it physically mean?

*We will expand this text as follows:* "The PCoA scores differentiate records geographically (Fig. 2a): southern hemisphere monsoon regions such as the southwestern South American Monsoon (SW-SAM) and South African Monsoon (SAfM) are characterised by low PCoA1 scores, whilst northern hemisphere monsoons such as the Indian Summer Monsoon (ISM) and the East Asian Monsoon (EAM), are characterised by higher PCoA1 scores. This indicates that regions can be differentiated based on their temporal evolution as captured by the first PCoA axis."

- l 300: "The regional composites are z-scores, i.e. anomalies with respect to the base period (3000-7000 yr BP)." Are these just anomalies or true z-scores? Please clarify how you calculate those z-scores and what they physically mean. And why using z-scores in the first place? Why not just simple anomalies?

*Speleothem $\delta^{18}O$ values are converted to z-scores when constructing regional composites as this method standardises both the mean and the variance (unlike anomalies). We will emphasise this in the text adding the equation for the calculation of z-scores in the methods section to make it clearer, as follows (from line 219):*

"The $\delta^{18}O$ data for individual speleothems were transformed to z-scores, so all records have a standardised mean and variance:

$$z\text{-}score_i = \left(\delta^{18}O_i - \overline{\delta^{18}O}_{(base\ period)}\right) / s\delta^{18}O_{(base\ period)}$$

Where $\overline{\delta^{18}O}$ is the mean and $s\delta^{18}O$ is the standard deviation of $\delta^{18}O$ for a common base period. A base period of 7,000 to 3,000 years BP was chosen to maximise the number of records included in each composite."

*In the results section 3.3 (from line 300) when describing what the z-scores show, we will reword to more clearly state that z-scores show a standardised mean and variance with respect to the base period. We believe this will address the reviewer's concerns by allowing readers to interpret the regional speleothem composites of fig 4:*

"The regional composites are expressed as z-scores, i.e. changes with respect to the mean and variance of $\delta^{18}O$ for the base period (3000-7000 yr BP)."

- Fig 4: what are the units of the plotted variables? Please add the units on the y-labels. I have trouble to compare the simulated and observed δ 18O: please use similar diagnostics and units for both. For example, convert precip δ 18O into calcite δ 18O for the model, and use simple δ 18O anomalies for the speleothem observations.

*We will add units to the axis labels of fig 4 (W m$^{-2}$ for insolation, ‰ for $\Delta\delta^{18}O_{precip}$, z-scores are unitless).*

*The goal of this figure is to compare the large-scale (regional) temporal trends in observed $\delta^{18}O_{spel}$ and simulated $\delta^{18}O_{precip}$, rather than to make a direct quantitative comparison. The z-scores used for the speleothem composite trends standardise the variance of the records and are unitless. Anomalies are used for simulated $\delta^{18}O_{precip}$ without a conversion to $\delta^{18}O_{calcite}$ as the latter would require*

*information on the cave temperature which could only be inferred using simulated air temperature, which in turn would add more uncertainty to the comparison.*

- Fig 6: can you explain better how these diagrams should be interpreted? What do they physically mean?

*We define partial residual plots in the methods section (line 263). However, we will modify the text (at line 324) to provide a physical interpretation of these plots:*

"The global model for the Holocene (1 to 9ka) $\delta^{18}O_{precip}$ trends has a pseudo-$R^2$ of 0.80 and shows statistically significant relationships between the anomalies in $\delta^{18}O_{precip}$ and anomalies in regional precipitation, temperature and surface wind direction (Table 3). The partial residual plots (Fig. 6) show there is a strong negative relationship with regional precipitation (t value = -8.75) and a strong positive relationship (t value = 8.03) with surface wind direction over the moisture source region, an index of changes in either source area or moisture pathway. This indicates that increases in regional precipitation alone will lead to a decrease in $\delta^{18}O$ while changes in source area/moisture pathway, in the absence of changes in other variables, will lead to a significant change in $\delta^{18}O$. The relationship with temperature over the moisture source region is weaker, but positive (t value = 2.05), i.e. an increase in temperature over the moisture source region will lead to an increase in $\delta^{18}O$ if there are no changes in other climate variables. Precipitation recycling is not significant in this global analysis."

- Fig 6, section 3.4: on which model was this regression analysis done? GISS or ECHAM? More generally, why doing each diagnostic with only one model? Why not doing each diagnostic with each model (when the period of interest is available), to assess the robustness of the results?

*As explained in our answer to major comment 1.2, there are only a few isotope-enabled palaeoclimate simulations, and they use different protocols even when they run simulations for the same time period. Thus, it is difficult to compare the simulations or assess their robustness because there are multiple possible causes for any differences between them. We use ECHAM for glacial-interglacial shifts and GISS for Holocene evolution. This has been clarified in our proposed amendment to the text. We will also amend the methods section describing our statistical analyses of simulated $\delta^{18}O$ and climate variables to clarify which models are being used. Amendments are for line 174:*

"We examined glacial-interglacial shifts in $\delta^{18}O_{spel}$ observations and in annual precipitation-weighted mean $\delta^{18}O_{precip}$ from ECHAM-wiso in regions influenced by the monsoon. We focus on regional differences between MH, LGM and LIG with respect to the present-day for speleothems or the control simulation experiment for model outputs."

*And line 238:*

"We investigate the drivers of regional $\delta^{18}O_{precip}$, and by extension $\delta^{18}O_{spel}$, through the Holocene using multiple linear regression (MLR) of annual precipitation-weighted mean $\delta^{18}O_{precip}$ anomalies and climate variables from GISS modelE-R. Climate variables were chosen to represent the four potential large-scale drivers of regional changes in the speleothem $\delta^{18}O$ records."

- l 380: "drivers" -> "meteorological variables". This is just a statistical analysis, so no causality can be identified, so the meteorological variables cannot be assumed to be drivers.

*Please see comment above related to this. We will change "drivers" to "climate variables"*

- l 389: "changes in precipitation amount" -> "changes in local precipitation amount": changes in upstream precipitation amount has been shown to be very important in previous studies (e.g. Battisti et al. (2014)) but were not analyzed here.

*Simulated precipitation changes in this study are regional averages over the monsoon regions, thus they are not equivalent to "local precipitation amount". To clarify this point, we will revise the occurrences of "precipitation amount" throughout to "regional precipitation", including at line 389:*

*"Changes in regional precipitation do not seem to explain the observed changes in $\delta^{18}Ospel$ in the EAM during the Holocene."*

- Table 1: too many digits in the numbers.

*We will reduce values in table 1 to two decimal places*

*Refs:*

Comas-Bru, L., Harrison, S. P., Werner, M., Rehfeld, K., Scroxton, N. and Veiga-Pires, C.: Evaluating model outputs using integrated global speleothem records of climate change since the last glacial, Clim. Past., 15(4), 1557-1579, https://doi.org/10.5194/cp-15-1557-2019, 2019.

Gierz, P., Werner, M. and Lohmann, G.: Simulating climate and stable water isotopes during the Last Interglacial using a coupled climate-isotope model, J. Adv. Model. Earth Syst., 9(5), 2027–2045, https://doi.org/10.1002/2017MS001056, 2017.

Hu, J., Emile-Geay, J., Tabor, C., Nusbaumer, J. and Partin, J.: Deciphering oxygen isotope records from Chinese speleothems with an isotope-enabed climate model, Paleoceanogr. Paleoclimatology, 43(12), 2098-2112, https://doi.org/ 10.1029/2019PA003741, 2019.

LeGrande, A. N. and Schmidt, G. A.: Sources of Holocene variability of oxygen isotopes in paleoclimate archives, Clim. Past, 5(3), 441–455, https://doi.org/10.5194/cp-5-441-2009, 2009.

Maher, B. A.: Holocene variability of the East Asian summer monsoon from Chinese cave records: A re-assessment, Holocene, 18(6), 861–866, https://doi.org/10.1177/0959683608095569, 2008.

Sinha, A., Kathayat, G., Cheng, H., Breitenbach, S.F.M., Berkelhammer, M., Mudelsee, M., Biswas, J. and Edwards, R.L.: Trends and oscillations in the Indian summer monsoon rainfall over the last two millennia, Nat. Commun., 6(6309), https://doi.org/10.1038/ncomms7309, 2015.

Ward, B., Wong, C., Novello, V., McGee, D., Santos, R.V., Silva, L.C.R., Cruz, F.W., Wang, X., Edwards, R.L. and Cheng, H.: Reconstruction of Holocene coupling between the South American Monsoon System and local moisture variability from speleothem $\delta^{18}O$ and $^{87}Sr/^{86}Sr$ records, Quat. Sci. Rev., 210, 51-63, http://doi.org/10.1016/j.quascirev.2019.02.019, 2019.

Wackerbarth, A., Langebroek, P. M., Werner, M., Lohmann, G., Riechelmann, S., Borsato, A. and Mangini, A.: Simulated oxygen isotopes in cave drip water and speleothem calcite in European caves, Clim. Past, 8(6), 1781–1799, https://doi.org/10.5194/cp-8-1781-2012, 2012.

Werner, M., Jouzel, J., Masson-Delmotte, V. and Lohmann, G.: Reconciling glacial Antarctic water stable isotopes with ice sheet topography and the isotopic paleothermometer, Nat. Commun., 9(1), 1–10, https://doi.org/10.1038/s41467-018-05430- y, 2018.

---

## Author Comment (AC2) · 9 Oct 2020

Response to referee #2

**Major comments**

1) The introduction does not reflect the current understanding of the speleothem δ18O, particularly in the East Asian monsoon domain. For example, it basically follows the previous misunderstanding(s) from modeling and other research communities, especially on orbital-scale, that the speleothem δ18O was interpreted as a rainfall amount proxy by the Chinese speleothem community over the past two decades. In fact, the mainstream idea from the speleothem community has never been the 'amount affect' (e.g., Cheng et al., 2019), and therefore, one of main scientific issues addressed here is groundless.

*We understand from the reviewer's concerns that the introduction requires rewording to make clear how Chinese speleothem records have been interpreted as monsoon signals in the literature, i.e. as an upstream precipitation signal (Hu et al., 2008; Yuan et al., 2004) or a rainfall seasonality signal (Cheng et al., 2006, 2009; Wang et al., 2001).Other interpretations of Chinese monsoon $\delta^{18}O_{spel}$ have included rainfall source changes (Tan 2009, 2011, 2014) or local rainfall changes in specific areas (Cai et al., 2010; Tan et al., 2015). Changes to the text to address this are proposed under comment 3.*

*However, we would like to clarify that our focus here is not simply on the East Asian monsoon domain but rather to investigate all regional monsoons with sufficient speleothem data available in the SISALv2 database. Our discussion in the introduction was to highlight the fact that multiple mechanisms have been proposed in the existing literature to explain $\delta^{18}O_{spel}$ trends in monsoon regions, and we use the East Asian monsoon region as one example of this. We also provide examples from other regions, including the Indonesian-Australian monsoon (line 74 et seq.) and the South American monsoon (line 76 et seq.). To better emphasise this point, we will expand our discussion of other regions in the introduction (under comment 3), rather than mostly discussing the interpretation of East Asian speleothems. It is not uncommon in the literature to propose only one (dominant) mechanism to explain $\delta^{18}O_{spel}$ variability when in reality there could be several mechanisms acting in combination. Furthermore, the proposed mechanisms are often based on modern-day observed relationships, which may not have remained constant in the past. In this study, we utilise model simulations that incorporate known isotope effects/physics, under considerably different conditions to today and we use multiple regression analysis to account for multiple possible isotope drivers in combination. We will clarify this point when describing the aims of this paper in the introduction as follows (after line 91):*

"In this study, we combine speleothem $\delta^{18}O$ records from version 2 of the Speleothem Isotopes Synthesis and Analysis (SISAL) database with isotope-enabled palaeoclimate simulations from two climate models to investigate the plausible mechanisms driving changes in $\delta^{18}O$ in monsoon regions through the Holocene (last 11,700 years) and between interglacial (mid-Holocene and Last Interglacial) and glacial (Last Glacial Maximum) states.

*And after line 98:* We use isotope-enabled model simulations to investigate the main drivers of $\delta^{18}O_{spel}$ variability in regions where the models reproduce the large-scale $\delta^{18}O$ changes shown by observations. We exploit the fact that models produce internally physically consistent changes to explore potential and plausible causes of the trends observed in speleothem records across specific monsoon regions, using multiple regression analysis."

2) The authors mentioned that "a composite record can minimize the influence of site specific karst and cave processes" (with real spatial variations?). However, the results and/or assumptions from the PCoA method are tentative, which lacks a underlying mechanism. The same monsoon system (e.g., the ISM and EAM boxes in the figure 1) could have different speleothem δ18O patterns on orbital-scale, as illustrated by a number of modeling results (e.g., Liu et al., 2014; Battisti et al., 2014).

*The reviewer has misunderstood the purpose of the PCoA analysis. We use PCoA to investigate the (dis)similarity of Holocene $\delta^{18}O_{spel}$ trends in order to be able to determine whether there is any large-scale coherency between individual monsoon speleothem records and thus whether it is possible to group records based on the similarity of their Holocene trends in a quantitative and objective way. By showing that records show geographic coherency, we are able to construct regional composites which we subsequently use to study mechanisms through multiple regression.*

*We will clarify the purpose of the PCoA by amending the text from line 268:*

*"PCoA shows the (dis)similarity of Holocene $\delta^{18}O_{spel}$ evolution across individual records, and thus allows an objective regionalisation of these records."*

3) Lines 66-68: This is really a misleading statement. I suggest that the authors should read the original papers they cited here more carefully (as well as Cheng et al., 2016, 2019; Zhang et al., 2018; Zhang et al., 2020) and quote the original statements in these papers if necessary. For example, Cheng et al. (2009) (cited in the sentence) clearly asserted: "Thus, neither the temperature- δ18O relationship, commonly used to interpret ice-core data, nor the interpretation based on the "amount effect" is justified".

*Our purpose here, as explained above, was simply to demonstrate that there are alternative interpretations of the records from specific regions rather than to review the literature from any one region exhaustively. However, we will expand this text to reflect what these various papers meant when discussing summer precipitation changes (from line 66):*

*"In the East Asian monsoon, for example, speleothem $\delta^{18}O$ records have been interpreted as a summer monsoon signal, manifested as either a change in the amount of water vapour removed along the precipitation trajectory (Yuan et al., 2004), and/or as a change in the contribution of summer precipitation to annual totals (Cheng et al., 2006, 2009, 2016; Wang et al., 2001), based on the relationship between modern $\delta^{18}O_{precip}$ and climate. Other interpretations of Chinese monsoon $\delta^{18}O_{spel}$ have included rainfall source changes (Tan 2009, 2011, 2014) or changes in monsoon precipitation amount (Cai et al., 2010; Tan et al., 2015). Maher (2008) interpreted $\delta^{18}O_{spel}$ as reflecting changes in moisture source area, based on differences between $\delta^{18}O_{spel}$ and loess/palaeosol records of rainfall and the strong correlation between East Asian and Indian monsoon speleothems. Maher and Thompson (2012) used a mass balance approach to show that the changes in precipitation (either local or upstream) or rainfall seasonality required to reproduce $\delta^{18}O_{spel}$ trends would be unreasonably large. They therefore argued that changes in moisture source were required to explain shifts in $\delta^{18}O$ both on glacial/interglacial time scales and during interglacials. There are also multiple interpretations of the causes of $\delta^{18}O_{spel}$ variability in other monsoon regions. In the Indonesian-Australian monsoon region, for example, $\delta^{18}O_{spel}$ variability has been interpreted as a precipitation amount signal (Carolin et al., 2016; Krause et al., 2019) or a precipitation seasonality signal (Ayliffe et al., 2013; Griffiths et al., 2009), based on modern $\delta^{18}O_{precip}$ and climate observations (Cobb et al., 2007; Moerman et al., 2013), and/or as a moisture*

source/trajectory signal (Griffiths et al., 2009; Wurtzel et al., 2018). South American speleothem records have been interpreted as records of monsoon intensity, due to changes in the amount of precipitation over the region (Cruz et al., 2006; Wang et al., 2006; Cheng et al., 2013), changes in the degree of upstream precipitation and evapotranspiration (Cheng et al., 2013) or changes in the ratio of precipitation sourced from the low-level jet versus the Atlantic (Cruz et al., 2005; Wang et al., 2006). In the Indian monsoon region, speleothem $\delta^{18}O$ records are interpreted primarily as an amount effect signal (Berkelhammer et al., 2010; Fleitmann et al., 2004), supported by $\delta^{18}O_{precip}$/climate observations (e.g. Battacharya et al., 2003). However, other studies have suggested that $\delta^{18}O_{precip}$ changes in this region are driven primarily by large-scale changes in monsoon circulation and hence, Indian monsoon $\delta^{18}O_{spel}$ should be interpreted as a moisture source/trajectory signal (Breitenbach et al., 2010; Sinha et al., 2015). "

4) Lines 229-236 and figure 4: What are the simulated precipitation δ18O values in the EAM, ISM, IAM, SW-SAM domains? Are they amount-weighted annual mean precipitation δ18O values, annual mean precipitation δ18O values or only summer (MJJAS) mean precipitation δ18O values? In addition, please give the boundary coordinate of these monsoon regions (the EAM, ISM, IAM, SW-SAM. . .) for the calculations. Give a detail explanation about the δ18O amplitude differences between observation and model results in the figure 4 if significant.

*All simulated $\delta^{18}O_{precip}$ values are annual precipitation-weighted $\delta^{18}O$ anomalies with respect to a control simulation. We will amend the text as follows:*

*Line 174:* "We examined glacial-interglacial shifts in $\delta^{18}O_{spel}$ observations and in annual precipitation-weighted mean $\delta^{18}O_{precip}$ from ECHAM-wiso in regions influenced by the monsoon. We focus on regional differences between MH, LGM and LIG with respect to the present-day for speleothems or the control simulation experiment for model outputs."

*Line 229:* "We calculated Holocene regional composites from annual precipitation-weighted mean $\delta^{18}O_{precip}$ anomalies simulated by the GISS model."

*Line 238:* "We investigate the drivers of regional $\delta^{18}O_{precip}$, and by extension $\delta^{18}O_{spel}$, through the Holocene using multiple linear regression (MLR) of annual precipitation-weighted mean $\delta^{18}O_{precip}$ anomalies and climate variables from GISS modelE-R. Climate variables were chosen to represent the four potential large-scale drivers of regional changes in the speleothem $\delta^{18}O$ records.

*We will add the latitude/longitude limits of the regional monsoons to the caption of figure 1:*

Figure 1: Spatial distribution of speleothem records used is this study. Colours indicate the sites used in Principal Coordinates Analysis and Redundancy Analysis (PCoA, RDA) to separate monsoon regions, and sites not used in PCoA and RDA but used in subsequent analyses. The individual regional monsoons are shown by boxes: CAM = Central American Monsoon (latitude: 10 to 33°; longitude: -115 to -58°) , SW-SAM = southwestern South American Monsoon (latitude: -10° to 0°; longitude: -80° to -64° and latitude: -30° to -10°; longitude -68° to -40°), NE-SAM = northeastern South American Monsoon (latitude: -10° to 0°; longitude: -60° to -30°), SAfM = southern African Monsoon (latitude: -30° to -17°; longitude: 10° to 40°), ISM = Indian Summer Monsoon (latitude: 11° to 32°; longitude: 50° to 95°), EAM = East Asian Monsoon (latitude: 20° to 39°; longitude: 100° to 125°), IAM = Indonesian-Australian Monsoon (latitude: -24° to 5°; longitude: 95° to 135°). Source region limits

used in the multiple linear regression analysis are also shown. The background carbonate lithology is from the World Karst Aquifer Mapping (WOKAM) project (Goldschneider et al., 2020).

5) Lines 376-379: ". . .there is little different in the δ18O values between the MH and the LIG in the ISM and EAM regions. . .", "Given that the increase in summer insolation is much larger during the LIG than the MH, this finding is again consistent with the idea that other factors play a role in modulating the monsoon response to insolation forcing". What are the other factors and the processes? Moisture source and/or pathway? Or some kind of thresholds (e.g., Cheng et al., 2012; Cai et al., 2015)? In addition, the summer insolation is indeed higher during the LIG than during the MH, but the monsoon circulation or intensity is influenced by the temperature (thus pressure?) gradient between land and sea as well. What is the difference of the land-sea temperature (pressure) gradients for the MH and the LIG periods? Or monsoon circulation scales? A more comprehensive discussion of the issue with a help of climate models would be very welcome.

*An in-depth discussion of the influences on the East Asian and Indian monsoons is beyond the scope of this paper, since it requires consideration of the monsoons as an integral part of the global atmospheric overturning circulation (see e.g. Schneider et al., 2014; Biasutti et al., 2018; Seth et al., 2019) and associated energy, angular momentum, and moisture budgets. Given that the monsoons cannot simply be considered as regional land-sea breeze circulations, analysis of the land-sea temperature/pressure gradients in the MH and LIG would be insufficient. Our point here was to support the idea, expressed in this paragraph, that there is no simple correspondence between insolation forcing and monsoon response. We have argued that land and ocean feedbacks might have played a role in modulating the response to insolation changes during the Holocene - and the pattern of change through the LIG would also support this. We will clarify our argument about the role of insolation on monsoon changes as follows (line 377):*

"The evolution of regional monsoons during the LIG shows patterns similar to those observed during the Holocene, including the lagged response to insolation and the persistence of wet conditions after peak insolation. This is again consistent with the idea that internal feedbacks play a role in modulating the monsoon response to insolation forcing. We have also shown that there is little difference in the isotopic values between the MH and the LIG in the ISM and EAM regions, which is also observed in individual speleothem records (Kathayat et al., 2016; Wang et al., 2008). Given that the increase in summer insolation is much larger during the LIG than the MH, this finding indicates that other factors play a role in modulating the monsoon response to insolation forcing and may reflect the importance of global constraints on the externally-forced expansion of the tropical circulation (Biasutti et al., 2018)."

6) The main conclusion is that "East Asian monsoon speleothem δ18O evolution through the Holocene relates to changes in atmospheric circulation (i.e. changes in moisture pathway and/or source). Changes in precipitation amount are the predominant driver of Holocene δ18Ospel evolution in the Indian, southwestern South American and Indonesian-Australian monsoons, although changes in atmospheric circulation also contribute in the Indian and Indonesian-Australian monsoon regions and changes in precipitation recycling in southwestern South America". This conclusion is not well supported and problematic as well. First, the 'amount effect' discussed here is not the same 'amount effect' as

conventionally defined in the tropics (see Zhao et al., 2019 for instance). The authors implies that the local rainfall amount drive the orbital-scale variations in speleothem δ18O value. They really need to provide a mechanism/calculation for the Indian, southwestern South American and Indonesian-Australian monsoon systems to explain how the oxygen isotopic fractionation under different conditions of rainfall amounts at each cave site could result in the observed δ18Ospel changes on orbital-scale without significant monsoon circulation (including the moisture pathway and/or source) changes. On the other hand, the "East Asian monsoon speleothem δ18O evolution through the Holocene relates to changes in atmospheric circulation" is just a reinforcement of the previous view on the East Asian monsoon evolution inferred by speleothem δ18O records published in a large number of speleothem works over the past two decades. In short, it is the monsoon circulation that to first order drives the orbital δ18Ospel changes, not only for the East Asian monsoon, but also (most likely) for other monsoon systems.

*The reviewer is correct that we are talking about regionally averaged precipitation changes rather than changes in what is conventionally understood as the precipitation amount effect and we will change this wording to "changes in regional precipitation". However, the reviewer has misunderstood the purpose of our analyses. We acknowledge that there may be concurrent changes in multiple factors. However, the aim of multivariate analysis is to separate out the various $\delta^{18}O_{precip}$-climate relationships and investigate which variables (it may be a combination of several) are most important in a given region. For example, if circulation and regional precipitation were changing in a way that both together can explain $\delta^{18}O_{precip}$ trends, the MLR model would show this. On the other hand, if there was a significant change in atmospheric circulation (and hence source area) without a corresponding change in regional precipitation, there would be conformity with the global circulation relationship, but not with precipitation, as is seen for the EAM. This would be possible if circulation changes drove a precipitation change outside of the region where the speleothem sites are located, as has been proposed by several papers focusing on the EAM (e.g. Hu et al., 2008; Liu et al., 2014). In all cases, these varying atmospheric and/or precipitation changes are underpinned by the physics incorporated in the climate model simulations. One point that the reviewer appears to have missed is that our aim is not to disprove or reinforce conclusions based on previous EAM speleothem studies. Rather we are using multivariate analysis to provide an alternative way of examining the potential causes of observed changes, independently from the assumption that underpins most interpretations in the literature that modern relationships provide a robust guide to what has happened in the past. By analysing the individual effects of different variables using isotope-enabled models that reproduce the large-scale monsoon trends shown by the observations, we are able to determine what factors are important in a robust way.*

*We will make the following amendments to the text in order to make the purpose of our analyses clearer:*

*In the introduction, we will reword to make clearer the goal of this study (under comment 1).*

*In the results, we will more clearly state what the multiple linear regression shows (line 319):*

"The MLR analyses of simulated $\delta^{18}O_{precip}$ trends identify the impact of an individual climate variable on $\delta^{18}O_{precip}$ in the absence of changes in other variables."

*When discussing the multiple linear regression (line 389):*

Changes in regional precipitation (where the cave sites are located) do not seem to explain the observed changes in $\delta^{18}O_{spel}$ in the EAM during the Holocene, where Holocene $\delta^{18}O_{precip}$ evolution is largely driven by changes in atmospheric circulation (indexed by changes in surface winds). This is consistent with existing studies that emphasise changes in moisture source and/or pathway rather than local precipitation changes (Maher, 2016; Maher and Thompson, 2012; Tan, 2014; Yang et al., 2014).

7) Please illustrate the x- and y-axes of the figure 2a in the section 3.1 or describe them in the section 2.3. In the section 3.1, the authors illustrated that Southern Hemisphere monsoon regions are characterized by low PCoA1 scores, while Northern Hemisphere monsoon regions are characterized by higher PCoA1 scores. Please explain these terms in the context of instrumental data or modern climatology, which may be more interesting for the paleoclimate community.

*The aim of the PCoA is to investigate the (dis)similarity of Holocene $\delta^{18}O_{spel}$ trends amongst speleothem sites. We then use RDA to investigate whether the distribution of site (dis)similarity relates to geographic location (latitude and longitude). This allows us to investigate whether there is a regional and global-scale coherency to Holocene $\delta^{18}O_{spel}$ records, and thus to regionalise the records based on the observations themselves rather than any assumption of regional synchroneity in the speleothem records. We make no assumption that these trends are related to modern climatology, or that regions should be defined on the basis of their modern climatology. We have modified our description of the purpose of the PCoA analyses (in response to comments from reviewer 1) and this will hopefully make the purpose of these analyses clearer.*

8) The authors used the anomaly for comparison from different model results. However, readers might also want to see a detailed comparison between model results, particularly between the model results from this study and those from previous studies.

*There are relatively few isotope-enabled palaeoclimate simulations, and they are generally run under different protocols/boundary conditions, thus precluding a rigorous comparison between them since it is difficult to attribute differences to model structure or experimental protocol. Furthermore, an analysis of model-based results per se is not the goal of this paper. In response to comments by reviewer 1, we have included anomaly maps of simulated $\delta^{18}O_{precip}$ from the simulations we are using for our analyses in the supplementary material.*

9) Lines 397-400 and the figure 3: "The LGM is characterised by a similar orbital configuration to today, however global ice volume was at a maximum and GHG concentrations were lower than present. The δ18Ospel anomalies are more positive during the LGM than the MH or LIG, suggesting drier conditions in the ISM, EAM and IAM, supported by simulated changes in δ18Ospel and precipitation (Fig. 3)." This sentence is again misleading. While the authors highlighted a similar orbital configuration between the LGM and today, they actually discussed the issue related to a comparison of the LGM with the MH or LIG, presumably implying that they have similar orbital configurations. The LGM (21±1ka) is near a Northern Hemisphere insolation minimum whereas the MH/LIG are near the insolation maxima. As such the related discussions should be rephrased, and so does the related conclusion, since the insolation difference should be taken into account together with GHG and the global ice

volume, because one could also argue that the δ18Ospel just follows the insolation with effect to a lesser extent from GHG and the global ice volume.

*We agree that it is not ideal to describe the LGM boundary conditions with respect to the modern day when the purpose of this paragraph is to contrast the LGM signals with those of the MH and LGM, so we will rephrase this to read (line 397):*

*The LGM is characterised by lower northern hemisphere summer insolation, globally cooler temperatures, expanded global ice volumes and lower GHG concentrations than either the MH or the LIG.*

*And rephrase the conclusion (line 405) as:*

*Enriched $\delta^{18}O_{precip}$ and $\delta^{18}O_{spel}$ values during the LGM must therefore be caused by a significant decrease in atmospheric moisture and precipitation that resulted from the cooler conditions.*

**Minor comments**

Lines 97, 112 and 160, 'the Principal Coordinate Analysis (PCoA)', the abbreviation occurred three times, keep the first one.

*We will amend the text so that PCoA is only defined at its first mention (at line 97).*

Line 121, please give the full name of the climate models: ECHAM5 and GISS E-R

*We will modify the text to define these acronyms as follows:*

*"Here we use simulations of opportunity from two isotope-enabled climate models: ECHAM5 (version 5 of the European Centre for medium range weather forecasting model in HAMburg) and GISS E-R (Goddard Institute for Space Studies Model version E-R)."*

Line 163, '. . .missing data that . . .' , 'that' should be 'than'?

*Yes, we will correct the text here.*

Line 189, what is the 'OIPC'?

*OIPC is the data set described in line 185-186. We apologise for not naming it there and will amend the text to do so, as follows: "... using as reference the Online Isotopes in Precipitation Calculator (OIPC: Bowen, 2018; Bowen and Revenaugh, 2003), a global gridded dataset of interpolated mean annual precipitation-weighted $\delta^{18}O_{precip}$ data."*

Lines 268-277, the abbreviations (EAM, SW-SAM, SAfM, CAM, IAM) occurred too late in the section 3.1, it's better put them in the introduction.

*The regional monsoons, and their abbreviations are not introduced until section 3.1, as the results from PCoA justify our grouping of the data in regional monsoons. We therefore introduce them here. However, abbreviations are also available in the caption of Figure 1, which is first cited in line 116.*

Line 358 'southern China Sea' should be 'South China Sea'.

*We will amend the text accordingly*

Figure 5, the time series for Dongge Cave can be replaced by a high-resolution timeseries, please double check with the database.

*We use speleothem records from the SISALv2 database because these have been standardised, quality-controlled, and the age models have been verified. The higher resolution records of the LIG from Dongge cave (Kelly et al., 2006) are not in the SISAL database.*

*Refs:*

[revised manuscript text omitted]

---

## Referee Report (RR1)

**2nd review of the article by Parker et al, CP**

December 19, 2020

The paper has been improved by the authors and is now clearer and more accessible. The additional figures in the supplementary material are helpful.

However, my major comment remains. I think that the wording could be modified to clarify the limitations of this study.

**1 Major comment: correlations do not always indicate causality, even within the world of a GCM**

Reading the response to reviewers, I understand that the authors want to document what climate variables are associated with changes in speleothem $\delta^{18}O$. Ideally, they would have used observations only, but they argue why they prefer to use a GCM: I think this an important argument that should be added in the introduction of the paper.

I agree that a GCM provides a physically consistent framework, where all climate variables are available for analysis. However, the world of he GCM is extremely complex, almost as complex as reality. Analyzing GCM outputs to identify drivers of $\delta^{18}O$ variations is thus extremely complex. This is why different authors in previous studies have developed decomposition methods to quantify the relative effects of different processes (e.g. [Botsyun et al., 2016, Tabor et al., 2018]). In absence of such decomposition methods, the drivers cannot be quantified. At best, you can look at how $\delta^{18}O$ variations correlate with climate variables. This is what you do. This identifies concomitant changes, but not drivers. Some concomitant variations may be fortuitous, mediated by other variables, or may contribute to a small fraction of $\delta^{18}O$ variations. I think this should be clarified in the paper. The wording "causes", "drivers", "explanations" should be avoided, for example:

- l 91: "provide plausible explanations for" -> "provide the changes in climate variables associated with"

- l 110: "main drivers of" -> "changes in climate variables associated with"

- l 112: "potential and plausible causes of" -> "trends in climate variables associated with"

- same l 232, 262, 467, 469, 475, ...

In the discussion and conclusion, the main limitations of the approach should be recalled: (1) limitations associated with the GCM-observation mismatches, emphasizing the need for a thorough evaluation of the GCM simulations; and (2) limitations related to the correlation analysis that does not allow to identify drivers: a decomposition method would be necessary.

**2 Minor comments**

- l 59: "reducing" -> "weakening" (because it's probably negative)

- l 345-347: "warmer and wetter": this would have opposite effects on $\delta^{18}O$. So what is $\delta^{18}O$ consistent with?
  Same for "cooler and drier"

- l 462: this number is a local recycling ratio. For $\delta^{18}O$, what matters is the total fraction of the precipitation that comes from continental recycling on any land grid box, and the number can be larger than 50% ([Yoshimura et al., 2004, Risi et al., 2013]). For the effect of continental recycling on paleo isotopic records, you may cite [Pierrehumbert, 1999].

- l 466: could the greater water-calcite fractionation at colder temperature also contribute to the observed change in speleothem $\delta^{18}O$? Could you do at least a simple back-of-the-envelope calculation to check this?

- Fig 3: I still find it very inconvenient to have a different axis for the model and observations. The figure allows us to see the sign of changes, but not the amplitudes. If the model capture the sign but not the amplitude, this is a very important information. So can you please use the same axis?

- Fig 3 caption: "shown" -> "show"

**References**

[Botsyun et al., 2016] Botsyun, S., Sepulchre, P., Risi, C., and Donnadieu, Y. (2016). Impacts of tibetan plateau uplift on atmospheric dynamics and associated precipitation $\delta$ 18 o. *Climate of the Past*, 12(6):1401–1420.

[Pierrehumbert, 1999] Pierrehumbert, R. T. (1999). Huascaran delta18O as an indicator of tropical climate during the Last Glacial Maximum. *Geophys. Res. Lett.*, 26:1345–1348.

[Risi et al., 2013] Risi, C., Noone, D., Frankenberg, C., and Worden, J. (2013). Role of continental recycling in intraseasonal variations of continental moisture as deduced from model simulations and water vapor isotopic measurements. *Water Resour. Res.*, 49:4136–4156, doi: 10.1002/wrcr.20312.

[Tabor et al., 2018] Tabor, C. R., Otto-Bliesner, B. L., Brady, E. C., Nusbaumer, J., Zhu, J., Erb, M. P., Wong, T. E., Liu, Z., and Noone, D. (2018). Interpreting precession-driven $\delta$18o variability in the south asian monsoon region. *Journal of Geophysical Research: Atmospheres*, 123(11):5927–5946.

[Yoshimura et al., 2004] Yoshimura, K., Oki, T., Ohte, N., and Kanae, S. (2004). Colored moisture analysis estimates of variations in 1998 asian monsoon water sources. *J. Meteor. Soc. Japan*, 82:1315–1329.

---

## Author Response (AR2)

The paper has been improved by the authors and is now clearer and more accessible. The additional figures in the supplementary material are helpful. However, my major comment remains. I think that the wording could be modified to clarify the limitations of this study.

*Major comment:*

Correlations do not always indicate causality, even within the world of a GCM. Reading the response to reviewers, I understand that the authors want to document what climate variables are associated with changes in speleothem δ18O. Ideally, they would have used observations only, but they argue why they prefer to use a GCM: I think this an important argument that should be added in the introduction of the paper. I agree that a GCM provides a physically consistent framework, where all climate variables are available for analysis. However, the world of the GCM is extremely complex, almost as complex as reality. Analyzing GCM outputs to identify drivers of δ18O variations is thus extremely complex. This is why different authors in previous studies have developed decomposition methods to quantify the relative effects of different processes (e.g. [Botsyun et al., 2016, Tabor et al., 2018]). In absence of such decomposition methods, the drivers cannot be quantified. At best, you can look at how δ18O variations correlate with climate variables. This is what you do. This identifies concomitant changes, but not drivers. Some concomitant variations may be fortuitous, mediated by other variables, or may contribute to a small fraction of δ18O variations. I think this should be clarified in the paper. The wording "causes", "drivers", "explanations" should be avoided, for example:

• l 91: "provide plausible explanations for" -> "provide the changes in climate variables associated with"

• l 110: "main drivers of" -> "changes in climate variables associated with"

• l 112: "potential and plausible causes of" -> "trends in climate variables associated with"

• same l 232, 262, 467, 469, 475

We agree with the referee that concomitant changes in $\delta^{18}O$ and meteorological variables does not necessarily mean that one drives the other. However, it is worth noting that the relationships identified in this paper between $\delta^{18}O$ and climate variables are consistent with our theoretical understanding of oxygen isotope systematics in the hydrological cycle and are consistent with other papers (as discussed from l417 to l433). Therefore, some of these terms: "plausible explanations", "potential and plausible causes" are suitable here. We will, however, reword several sentences that the referee suggests:

From l109:

"We use isotope-enabled model simulations, to investigate the potential causes of $\delta^{18}O_{spel}$ variability in regions where the models reproduce the large-scale $\delta^{18}O$ changes shown by observations."

From l232:

"We used anomalies of w$\delta^{18}O_{precip}$, mean annual surface air temperature (MAT) and mean annual precipitation (MAP) from the ECHAM5-wiso simulations to investigate the changes in $\delta^{18}O_{spel}$ between the MH, LGM and LIG, and their association with changes in climate."

From l262:

"We investigate the underlying relationships between regional $\delta^{18}O_{precip}$ (and by extension $\delta^{18}O_{spel}$) and monsoon climate through the Holocene using multiple linear regression (MLR)".

From l467:

"This study illustrates a novel data-model approach to investigate the relationship between $\delta^{18}O_{spel}$ and monsoon climate under past conditions: We compare composite regional records and then use multiple linear regression of isotope-enabled palaeoclimate simulations to determine the change in individual climate variables associated with these trends."

From l474:

"Geographically distributed speleothem $\delta^{18}O$ records and isotope-enabled climate models can be used together to understand the underlying relationships between $\delta^{18}O_{spel}$ and monsoon climate in the past and therefore elucidate possible drivers of $\delta^{18}O$ variability."

In the discussion and conclusion, the main limitations of the approach should be recalled: (1) limitations associated with the GCM-observation mismatches, emphasizing the need for a thorough evaluation of the GCM simulations; and (2) limitations related to the correlation analysis that does not allow to identify drivers: a decomposition method would be necessary.

We will add this point when discussing the limitations in this study.

From l458:

"Isotope-enabled climate models are used in this study to explore observed regional-scale trends in δ18Ospel. There is a limited number of isotope-enabled models, and there are no simulations of the same time period using the same experimental protocol. Although there are simulations of the MH from both ECHAM5-wiso and GISS, for example, these models have different grid resolutions and used different boundary conditions. This could help to explain why the two models yield different estimates of the change in regional δ18Oprecip (of 0.5 ‰) at the MH. However, both models show trends in δ18Oprecip that reproduce the observed changes in regional δ18Ospel (Figs 3 and 4), and this provides a basis for using these models to explore the potential causes of these trends on different timescales. The failure to reproduce the LGM δ18Ospel signal in SW-SAM in the ECHAM5-wiso model, which precluded a consideration of interglacial-glacial shifts in this region, is a common feature of other isotope-enabled simulations (Caley et al., 2014; Risi et al., 2010). Identifying the underlying relationships between $\delta^{18}O_{precip}$ and monsoon climate variables using multiple linear regression allows us to identify plausible mechanistic controls on $\delta^{18}O$ variability in the monsoon regions. Correlations between $\delta^{18}O$ and specific climate variables do not explicitly indicate causality. However, the relationships identified in the MLR model are consistent with the theoretical understanding of oxygen isotope systematics, and the findings of this paper are consistent with existing studies, suggesting that these relationships provide a plausible explanation for observed changes."

*Minor comments*

• l 59: "reducing" -> "weakening" (because it's probably negative)

We will amend the text as follows: "thereby minimising the $\delta^{18}O_{precip}$/distance gradient along an advection path…"

• l 345-347: "warmer and wetter": this would have opposite effects on δ 18O. So what is δ 18O consistent with? Same for "cooler and drier"

In this analysis, periods of more negative $\delta^{18}O$ values are warmer and wetter, whilst the LGM has less negative $\delta^{18}O$ values and cooler and drier conditions. The observation made in this section is that shifts are seen in all three variables. In the discussion (l434-l443), we elaborate that overall cooler conditions during the LGM result in a significant decrease in atmospheric moisture and therefore precipitation. Our use of the word "consistent" is unsuitable here, as we do not elaborate until later in the paper. Therefore, we will amend the text (at l320-l322) as follows:

"Glacial-interglacial shifts are also seen in precipitation and temperature, with warmer and wetter conditions during interglacials and cooler and drier conditions during the LGM in all three regions."

• l 462: this number is a local recycling ratio. For δ 18O, what matters is the total fraction of the precipitation that comes from continental recycling on any land grid box, and the number can be larger than 50% ([Yoshimura et al., 2004, Risi et al., 2013]). For the effect of continental recycling on paleo isotopic records, you may cite [Pierrehumbert, 1999].

Recycling in this paper is approximated using regional water budgets, i.e. the proportion of moisture derived from within the region versus the proportion of advected moisture, rather than oceanic versus terrestrial moisture sources. It therefore seemed appropriate to compare our findings with recycling values that were calculated the same way (Brubaker et al., 1993; Eltahir and Bras, 1994). However, we will add a sentence adding that recycling estimates derived from water tagging (Risi et al., 2013; Yoshimura et al., 2004) are higher still.

From l431:

"However, this is a region where changes in precipitation recycling also appear to be important. Based on regional water budget estimates, recycling presently contributes ca 25-35% of the precipitation over the Amazon (Brubaker et al., 1993; Eltahir and Bras, 1994) while these figures increase up to ca 40-60% based on moisture tagging studies (Risi et al., 2013; Yoshimura et al., 2004)."

• l 466: could the greater water-calcite fractionation at colder temperature also contribute to the observed change in speleothem δ 18O? Could you do at least a simple back-of-the-envelope calculation to check this?

Using equations for drip water to calcite (Tremaine et al., 2011) or aragonite (Grossman and Ku, 1986) oxygen isotope fractionation, the ~2°C cooler LGM temperatures for the ISM and EAM would cause $\delta^{18}O_{spel}$ to be ~0.4‰ more depleted and ~3°C cooling for the IAM results in $\delta^{18}O$ values ~0.6‰ more depleted. This will partially counteract the relatively more enriched $\delta^{18}O_{spel}$ values of the LGM, although the effect of cave temperature on $\delta^{18}O_{spel}$ is within the uncertainty of regional signals. We will amend the text to include these back-of-the-envelope calculations.

From l437:

"Cooler SSTs of approximately 2°C (relative to the MH and LIG) in the ISM and EAM and of approximately 3°C in IAM source areas, together with a ca 5% decrease in relative humidity (Yue et al., 2011) would result in a water vapour $\delta^{18}O$ signal at the source ca 1 ‰ more depleted than seawater. This depletion results from the temperature dependence of equilibrium fractionation during evaporation and kinetic isotope effects related to humidity (Clark and Fritz, 1997). This fractionation counteracts any impact from enriched seawater $\delta^{18}O$ values during the LGM (ca. +1 ‰ relative to the MH or LIG; Waelbroeck et al., 2002). Cooler air temperatures will also result in a depletion of $\delta^{18}O_{spel}$ values during the LGM of ca 0.4 ‰ and 0.6‰ for the ISM/EAM and IAM respectively, as a result of water-calcite/aragonite fractionation (Grossman and Ku, 1986; Tremaine et al., 2011). This has the effect of slightly reducing the regional LGM $\delta^{18}O_{spel}$ signals, although the change is small and within the uncertainty of the regional signals. The enriched $\delta^{18}O_{precip}$ and $\delta^{18}O_{spel}$ values during the LGM must therefore be caused by a significant decrease in atmospheric moisture and precipitation that resulted from the cooler conditions."

• Fig 3: I still find it very inconvenient to have a different axis for the model and observations. The figure allows us to see the sign of changes, but not the amplitudes. If the model capture the sign but not the amplitude, this is a very important information. So can you please use the same axis?

We reiterate that the key point of this figure is to compare the relative shifts of $\delta^{18}O$ and climate variables between the MH, LGM and LIG and we therefore fix the axes. However, we agree that comparing the amplitude and shifts of change between model and observation is interesting of itself. Therefore, we will include a figure of $\delta^{18}O_{precip}$ and $\delta^{18}O_{spel}$ (converted to $\delta^{18}O_{drip\ water}$), on the same axis, in the supplementary materials. It is worth noting the caveat that this necessary conversion of $\delta^{18}O_{spel}$ to $\delta^{18}O_{drip\ water}$ requires temperature values. We use modelled temperatures here, which adds a further source of uncertainty to $\delta^{18}O$ values.

We will add the following figure and caption to the supplement:

[Figure]

Figure S8: Speleothem $\delta^{18}O$ anomalies, converted to their drip water equivalent, compared to anomalies of $\delta^{18}O_{precip}$ from the ECHAM simulations for the (a) East Asian (EAM), (b) Indian (ISM) and (c) Indonesian-Australian (IAM) monsoons. $\delta^{18}O_{spel}$ (PBD) is converted to $\delta^{18}O_{drip water}$ (SMOW) following the methodology in Comas-Bru et al. (2019), using simulated temperature. The boxes show the median value (line) and the interquartile range, and the whiskers shown the minimum and maximum values, with outliers represented by grey dots. Note that the isotope axes are reversed, so that the most negative anomalies are at the top of the plot, to be consistent with the assumed relationship with the direction of change in precipitation and temperature. The difference in amplitude of $\delta^{18}O$ signals is small (<0.5 ‰) between simulated and observed values for the ISM and IAM. In the EAM, simulated $\delta^{18}O$ values have a ~1.4 ‰ higher amplitude than $\delta^{18}O_{drip water}$ observations.

We will add the following text, from l313:

"The simulated $\delta^{18}O_{precip}$ anomalies show a similar pattern, with more positive anomalies during the LGM than during the MH or the LIG. The amplitude of this pattern is also similar between $\delta^{18}O_{precip}$ and $\delta^{18}O_{spel}$, when the observations are converted to their drip water equivalent (Fig. S8)."

• Fig 3 caption: "shown" -> "show"

We will amend the text accordingly.

**Anonymous referee #2**

The revision has considerably improved the manuscript. I only have some minor comments/suggestions.

(1) Lines 62-63: "a change in the absolute amount of precipitation (Cai et al., 2012; Cheng et al., 2006)" is clearly a misinterpretation of the cited papers.

We apologise for the confusion here. We amended the text but failed to amend the citations appropriately. We will correct this mistake. We also acknowledge that the description of interpretations of the Chinese speleothem record in the introduction was over-simplified. We will amend the text to include the more complex interpretations by several papers:

From l61:

"Speleothem $\delta^{18}O$ records from monsoon regions show multi-millennial variability that has been interpreted as documenting the waxing and waning of the monsoons in response to changes in summer insolation, often interpreted predominantly as a change in the absolute amount of precipitation (Cheng et al., 2013; Fleitmann et al., 2003) or a change in the ratio of more negative $\delta^{18}O$ summer precipitation to less negative $\delta^{18}O$ winter precipitation (Dong et al., 2010; Wang et al., 2001). However, the multiplicity of processes that influence $\delta^{18}O$ before incorporation in the speleothem make it difficult to attribute the climatic causes of changes in individual speleothem records unambiguously. In the East Asian monsoon, for example, speleothem $\delta^{18}O$ records have been interpreted as a summer monsoon signal, manifested as a change in the amount of water vapour removed along the moisture trajectory (Yuan et al., 2004), and/or as a change in the contribution of summer precipitation to annual totals (Cheng et al., 2006, 2009, 2016; Wang et al., 2001) based on the relationship between modern $\delta^{18}O_{precip}$ and climate. Other interpretations of Chinese monsoon $\delta^{18}O_{spel}$ have included rainfall source changes (Tan 2009, 2011, 2014) or local rainfall changes (Tan et al., 2015). Maher (2008) interpreted $\delta^{18}O_{spel}$ as reflecting changes in moisture source area, based on differences between $\delta^{18}O_{spel}$ and loess/palaeosol records of rainfall and the strong correlation between East Asian and Indian monsoon speleothems. Maher and Thompson (2012) used a mass balance approach to show that the changes in precipitation (either local or upstream) or rainfall seasonality required to reproduce $\delta^{18}O_{spel}$ trends would be unreasonably large. They therefore argued that changes in moisture source were required to explain shifts in $\delta^{18}O$ both on glacial/interglacial time scales and during interglacials. Overall, there are several plausible climate mechanisms that could contribute to $\delta^{18}O_{spel}$ on multi-millennial timescales. East Asian monsoon speleothem records are often interpreted as a combination of several of these processes (Cheng et al., 2016; Dykoski et al., 2005) which overall represent monsoon intensity (Cheng et al., 2019)."

(2) In order to further demonstrate the various interpretations of Chinese speleothem δ18O records, it would be better to cite the recent review paper (Cheng et al., 2019. Chinese

stalagmite paleoclimate researches: A review and perspective. Science China: Earth Sciences 62 (10), 1489-1513), as suggested previously.

We will expand our discussion of East Asian monsoon $\delta^{18}O_{spel}$ in the introduction, to elaborate that interpretations often include multiple mechanisms, with $\delta^{18}O_{spel}$ representing an integrated result of processes, representing monsoon intensity, citing the Cheng et al. (2019) paper. We will amend the text as follows (from l76):

"East Asian monsoon speleothem records are often interpreted as a combination of several of these processes (Cheng et al., 2016; Dykoski et al., 2005) which overall represent monsoon intensity (Cheng et al., 2019)."

(3) It would be proper to move the description of the δ18Ospel in the Indian summer monsoon region (in lines 84-88) to line 77 (after the description about the East Asian monsoon), because both Indian summer monsoon and East Asian summer monsoon are a subsystem of the Asian summer monsoon that show strong interplays.

We will reorder this part of text to make the order more logical. From l76:

"There are also multiple interpretations of the causes of $\delta^{18}O_{spel}$ variability in other monsoon regions. In the Indian monsoon region, speleothem $\delta^{18}O$ records are interpreted primarily as an amount effect signal (Berkelhammer et al., 2010; Fleitmann et al., 2004), supported by $\delta^{18}O_{precip}$/climate observations (e.g. Battacharya et al., 2003). However, other studies have suggested that $\delta^{18}O_{precip}$ changes in this region are driven primarily by large-scale changes in monsoon circulation and hence, Indian monsoon $\delta^{18}O_{spel}$ should be interpreted as a moisture source/trajectory signal (Breitenbach et al., 2010; Sinha et al., 2015). In the Indonesian-Australian monsoon region, $\delta^{18}O_{spel}$ variability has been interpreted as a precipitation amount signal (Carolin et al., 2016; Krause et al., 2019) or a precipitation seasonality signal (Ayliffe et al., 2013; Griffiths et al., 2009), based on modern $\delta^{18}O_{precip}$ and climate observations (Cobb et al., 2007; Moerman et al., 2013), and/or as a moisture source/trajectory signal (Griffiths et al., 2009; Wurtzel et al., 2018). South American speleothem records have been interpreted as records of monsoon intensity, due to changes in the amount of precipitation over the region (Cruz et al., 2005; Wang et al., 2006; Cheng et al., 2013), changes in the degree of upstream precipitation and evapotranspiration (Cheng et al., 2013) or changes in the ratio of precipitation sourced from the low-level jet versus the Atlantic (Cruz et al., 2006; Wang et al., 2006)."

(4) The authors' response to the previous comment 9 is oversimply. Regarding the Asian monsoon, the LGM (~21 ±1ka) is under the condition of a Northern Hemisphere insolation minimum, whereas the MH and LIG are at the condition of an insolation maxima. This may at least partially explain the δ18Ospel records. More broadly, the heavy δ18Ospel excursions also occurred, for example, during the MIS 5b and 5d in the interglacial period (MIS 5) with similar amplitudes (values) to that of the LGM (e.g., Cheng et al., 2016. Nature). This is also the case in the South American monsoon regime, where the LGM δ18Ospel value is much lighter than those in MIS 5c (e.g., Cruz et al., 2005. Nature), and apparently these observations contradict in principle, to the current interpretation.

In this study, we focus on MIS 5e, i.e. the peak of the Last Interglacial, for comparison with the mid-Holocene and because this is the time period for which we have simulations. An examination of the variability within the stage 5 interglacial is beyond the scope of the paper. Similarly, we do not examine the entirety of the last Glacial period, but focus of the

LGM, when global ice volumes were at a maximum. We will amend the text to better clarify that we are focusing on the peak of the Last Interglacial (Stage 5e).

From l16-l18 (abstract):
"We focus on differences in $\delta^{18}O$ signals between the mid-Holocene, the peak of the Last Interglacial (Stage 5e) and Last Glacial Maximum, and on $\delta^{18}O$ evolution through the Holocene."

From l103-l105 (introduction):
"…to investigate the plausible mechanisms driving changes in $\delta^{18}O$ in monsoon regions through the Holocene period (last 11,700 years) and between the mid-Holocene, the peak of the Last Interglacial (Stage 5e) and Last Glacial Maximum."

From l121 (methods):
"For the analysis of mid-Holocene (MH), Last Glacial Maximum (LGM) and Stage 5e during the Last Interglacial (LIG) $\delta^{18}O$ signals, the records contain samples within at least one of these time periods, defined as 6,000±500 years for the MH, 21,000±1,000 years BP for the LGM and 125,000±1,000 years BP for the LIG, where BP (before present) is 1950 CE;

From l194 (methods):
"2.4. Glacial-interglacial changes in $\delta^{18}O$
We examined shifts in $\delta^{18}O_{spel}$ observations and in annual precipitation-weighted mean $\delta^{18}O_{precip}$ from ECHAM-wiso in regions influenced by the monsoon, between the MH, LGM and LIG. Values are given as anomalies with respect to the present day for speleothems and the control simulation for model outputs."

From l305 (results):
"3.2 Regional interglacial-glacial differences
To investigate the causes of shifts in $\delta^{18}O$ between the MH, LGM and LIG, we compare simulated and observed regional $\delta^{18}O$ signals during these periods with shifts in climate variables (precipitation and temperature)."

From l477 (conclusions):
"LGM $\delta^{18}O_{spel}$ signals are best explained by a large decrease in precipitation, as a consequence of lower atmospheric moisture content driven by global cooling."

We will also amend the caption of figure 5 to clarify that we are only examining variability within stage 5e, not the entire interglacial period. We will annotate figure 5 to mark the LIG (stage 5e) time slice used in the section 2.4 analysis:

[Figure]

"Figure 5: Comparison of changes in summer insolation and δ¹⁸O$_{spel}$ through the peak of the Last Interglacial (Stage 5e) from the (b,c) East Asian Monsoon (EAM), (d,e) Indian Summer Monsoon (ISM), (g) southwest South American Monsoon (SW-SAM) and (h) Indonesian-Australian Monsoon (IAM) regions. The U/Th dates and uncertainties are shown for each record. The summer insolation curves (Berger, 1978) are for May through September at 30° N in the northern hemisphere (a) and for November through March for 20° S in the southern hemisphere (f). Note that the isotope axes are reversed, so that the most negative anomalies are at the top of the plot, to be consistent with the assumed relationship with the changes in insolation. The LIG (Stage 5e) time slice used in the analysis in section 2.4 is shown by the dark grey bar."

**Steven Clemens**

The manuscript has improved considerably with the revisions in response to the two previous reviewers. It now reads more easily for those not familiar with the analytical approaches used. The authors utilize SISAL proxy data to define monsoon regions with similar speleothem d18O responses and then interrogate isotope enabled models to explain the patterns in the context of climate variables. The findings lend strong support to the evolving consensus that there is significant variability in the speleothem d18O response of the various monsoon systems and variability in the components of the climate system that drive the regional speleothem d18O signals (source area variability, dynamics along the moisture path, precipitation…).

Most of the comments below are largely along the lines of clarifications. Two can be considered more important, including the discussion of glacial-interglacial variability and the definition of source

areas, both of which, in my opinion, can be addressed at the discretion of the authors; I recommend publication with minor revision.

Title:
I recommend dropping '….on orbital timescales' from the title. The three short-duration time slices MH, LIG, and LGM explored here don't address the range of dynamics of orbital-scale variability expressed in paleoclimate records. More importantly, dropping these three words makes the title broader, better reflecting the idea that this approach can be applied to any time-scale of variability.
We agree that this paper does not fully explore orbital timescales. We will therefore change the title of the paper to:
"A data-model approach to interpreting speleothem oxygen isotope records from monsoon regions"

Abstract:
"Differences in speleothem δ18O between the mid-Holocene and Last Interglacial in the East Asian and Indian monsoons are small, despite the larger summer insolation values during the Last Interglacial."

Might this be due to the fact that CO2 is more similar in the mid-Holocene and present, implicating an internal radiative forcing versus an external insolation forcing? CO2 as a driver for East Asian monsoon variability is supported by most (all?) non-speleothem-based East Asian summer monsoon proxies examined at orbital time scales. See, for example, the Pleistocene loess proxies.
The difference in $CO_2$ concentrations is small between these two periods (LIG is 11 ppm higher; Otto-Bliesner et al., 2017). Whilst loess proxies at the margin of the East Asian monsoon show a close relationship between high northern latitude $CO_2$ concentrations and ice volumes (Lu et al., 2013; Sun et al., 2015), it is likely that the core monsoon region (where most speleothem sites are located) is less sensitive to these changes (Sun et al., 2015). Regardless, loess records suggest the East Asian monsoon is stronger when $CO_2$ concentrations are higher and ice volumes are lower. On this basis, higher $CO_2$ concentrations during the LIG would reinforce the enhancement of the monsoon by summer insolation and therefore cannot easily explain the similarity of $\delta^{18}O_{spel}$ signals between these two periods. We will amend the text to discuss $CO_2$ and ice volume difference.
From l l411:
"We have also shown that there is little difference in the isotopic values between the MH and the LIG in the ISM and EAM regions, which is also observed in individual speleothem records (Kathayat et al., 2016; Wang et al., 2008). The LIG (125ka) period was defined by higher summer insolation, higher $CO_2$ concentrations (Otto-Bliesner et al., 2017) and lower ice volumes (Dutton and Lambeck, 2012) than the MH, suggesting that the LIG ISM and EAM monsoons should be stronger than the MH monsoons. The lack of a clear differentiation in the isotope signals between the LIG and MH suggests that other factors play a role in modulating the monsoon response to these forcings and may reflect the importance of global constraints on the externally-forced expansion of the tropical circulation (Biasutti et al., 2018)."

1. Introduction:

Paragraph on isotope-enabled modeling (lines 93-105).

Jalihal et al., 2019 is a useful reference here as well (https://doi.org/10.5194/cp-15-449-2019).7

The Jalihal et al. (2019) is an interesting study of the complexity of influences on monsoon precipitation changes. However, in this paragraph we are focusing explicitly on results from isotope-enabled modelling - largely to provide the background to our use of isotope-enabled models in our analyses. Summarising the literature on the various influences on monsoon precipitation changes would be a much bigger task and not strictly relevant to our study.
* * *
2.6 Multiple regression analysis

"We investigate the drivers of regional δ18Oprecip, and by extension δ18Ospel, through the Holocene using multiple linear regression (MLR) of annual precipitation-weighted mean δ18Oprecip anomalies and climate variables from GISS model E-R. Climate variables were chosen to represent the four potential large-scale drivers of regional changes in the speleothem δ18O records. Specifically, we use changes in mean precipitation and precipitation recycling over the monsoon regions, and changes in mean surface air temperature and surface wind direction over the moisture source regions. Whereas the influence of changes in precipitation, recycling and temperature are relatively direct measures, the change in surface wind direction over the moisture source region is used as an index of potential changes in the moisture source region and transport pathway."

The three climate variables chosen are useful (and if recycling had not been presented, reviewers would have requested it). Nevertheless, I wonder if Including some variable that monitors changes in tropical deep convective precipitation (e.g. ISM) versus subtropical precipitation that is frontal in nature (e.g. EAM) could be useful in the MRL as well.

We agree that it would be worthwhile to investigate the role of deep convection. Unfortunately, the only variable we have that provides a measure of convection is convective precipitation. We calculated the relative proportion of convective precipitation (as the ratio of convective: total precipitation). However, both this ratio and convective precipitation itself are highly and significantly (P <0.001) correlated with precipitation (with correlation coefficients of 0.87 and 0.97 respectively) and therefore cannot be included in the model. Thus, including convective precipitation in the model in addition would not provide any additional useful information.
* * *
All climate variables were extracted for the summer months, defined as May to September (MJJAS) for northern hemisphere regions and November to March (NDJFM) for southern hemisphere regions (Wang and Ding, 2008).

This appears to be in conflict with the assumption (line 231-232) that "(i) precipitation-weighted mean annual δ18Oprecip is equivalent to mean annual drip-water δ18O (Yonge et al., 1985)". Please clarify; if the proxy reflects mean annual conditions, why compare to summer – season climate variables?

Speleothem $\delta^{18}O$ records are biased to the months when rainfall is highest, hence we use precipitation-weighted mean $\delta^{18}O_{precip}$ values. For most sites used in this study, there is a clear summer precipitation maximum; most are located within monsoon regions (defined by Wang and Ding, 2008) and hence are dominated by summer precipitation. We therefore use the summer mean for meteorological values.

We will add the following figure and caption to the supplement:

[Figure]

Figure S7: Spatial distribution of speleothem sites used in this study, shown with monsoon regions, defined by the precipitation-based criteria (Wang and Ding, 2008): the annual precipitation range (summer minus winter) exceeds 300 mm and 50 % of the annual mean. Summer is defined as May to September for the northern hemisphere and November to March for the southern hemisphere, vice versa for winter. Precipitation data is from the WFDEI dataset (Weedon et al., 2014).

We will amend the text from l272:
"All climate variables were extracted for the summer months, defined as May to September (MJJAS) for northern hemisphere regions and November to March (NDJFM) for southern hemisphere regions (Wang and Ding, 2008), on the basis that these regions are dominated by summer season precipitation (Fig. S7)."
* * *
3.2 Regional interglacial-glacial differences
"To investigate the causes of glacial-interglacial shifts in δ18O, we compare simulated and observed regional δ18O signals during the LIG, LGM and MH with shifts in climate variables (precipitation and temperature). Only the ISM, EAM and IAM regions have sufficient speleothem data (i.e. at least one record from every time period) to allow comparisons across the MH, LGM and LIG (Fig. 3) and have similar shifts in observed δ18Ospel and simulated δ18Oprecip. T. ….. The most positive δ18Ospel anomalies in all three regions occur at the LGM, with more negative anomalies for the MH and LIG."

A hallmark of the EAM orbital-scale composite speleothem record from the Yangtze River Valley is that it has no 100-kyr glacial-interglacial variance in the spectrum; it's virtually all precession. Discussion of termination occurrences in Cheng et al., is in the context of one every 4 or 5 precession cycles. Hence, this section (and associated figure 3 and discussion), is interpreting EAM precession-scale variance as glacial-interglacial (100-kyr) variance. In contrast, the ISM record from Xiaobialong (XBL) contains "real (100-ky)" glacial-intgerglacial variability and the associated publication (Cai et al., 15) discusses it in the context of global ice volume and sea level change. In this context, the proxy-model comparison result that "The glacial-interglacial changes in δ18Oprecip are consistent with the simulated temperature and precipitation changes, with warmer and wetter conditions during interglacials and cooler and drier conditions during the LGM in all three regions'' may be valid for the ISM but not the EAM. The EAM differences between values at the time of the LGM and the times of the MH and LIG may be better interpreted in the context of precession-band differences in model results (precession maxima vs precession minima) instead of LGM vs MH or LIG. I guess the point is that it's odd to discuss glacial-interglacial differences for records that have no 100-kyr variance.

For this analysis, we focus on the MH and LIG to represent interglacial peaks, whilst the LGM represents a period of maximum ice extent during the Last Glacial Period. Hence, this analysis provides snapshots of glacial-interglacial variability, but not a fuller picture capturing multiple terminations. We will clarify that the LIG here represents Stage 5e. Furthermore, we will further clarify the aim of this analysis in the text.

From l434:

"The LGM is characterised by lower northern hemisphere summer insolation, globally cooler temperatures, expanded global ice volumes and lower GHG concentrations than either the MH or the LIG. The MH and LIG (Stage 5e) periods represent peaks in the present and last interglacial periods, whilst the LGM represents maximum ice extent during the Last Glacial Period. Hence, comparison of these time periods provides a snap-shot view of glacial-interglacial variability. The $\delta^{18}O_{spel}$ anomalies are more positive during the LGM than the MH or LIG, suggesting drier conditions in the ISM, EAM and IAM, supported by simulated changes in $\delta^{18}O_{precip}$ and precipitation (Fig. 3)."
* * *
Figure 1.

The ISM source region seems too small if it includes only the red box encompassing the Arabian Sea; Indian summer monsoon moisture is sourced on the order of 50% from monsoon lows and depressions originating in the Bay of Bengal and tracking NW into India. Please clarify if the Bay of Bengal is or is not included as possible source region for the ISM and if not, explain why.

We defined monsoon regions based on modern moisture tracking studies (described from l270-272) and modelled surface winds. We also kept all source regions broadly a similar size. We originally tested several different source areas for each region, finding that our extracted values were not sensitive to the exact choice of source region limits.

We agree that Bay of Bengal is also an important moisture source for the ISM. However, expanding the ISM source region to include the Bay of Bengal as well as the Arabian Sea has a negligible effect on the results of the multiple linear regression model. We will include the figure and table for the MLR analysis for when this expanded source region is used in the supplement, and amend the text.

From l357:

"The exact choice of source region has a negligible impact on the model, for example expanding the ISM source region to include the Bay of Bengal does not change the outcome of this analysis (Fig S9, Table S1)."

In the supplement:

[Figure]

Figure S9: Same as figure 6, except Indian Summer Monsoon (ISM) source region has been expanded to include the Bay of Bengal.

| | Regression coefficient | T value |
|---|---|---|
| Regional precipitation | **-0.94** | -11.22 |
| Source area temperature | **0.52** | 2.95 |
| Wind direction | **0.06** | 8.30 |
| Precipitation recycling | 4.32 | 1.95 |

Table S1: Results of the multiple linear regression analysis when the Indian monsoon source region is expanded to include the Bay of Bengal. Significant relationships (P > 0.01) are shown in bold.
* * *
3.4 Multiple regression analysis of Holocene δ18Oprecip
"The global model for the Holocene (1 to 9ka) δ18Oprecip trends has a pseudo-R2 of 0.80 and shows statistically significant relationships between the anomalies in δ18Oprecip and anomalies in regional precipitation, temperature and surface wind direction (Table 3)."

The term 'global model' is not defined anywhere in the paper and, hence, confusing. Please clarify what model is used and if the results apply to the entire globe (pole to pole) or all monsoon regions combined…
The "global model" refers to the incorporation of all monsoon regions. We understand that this wording is potentially misleading and will therefore amend the text to clarify how the model is constructed.
From l262 (section 2.6):

"We investigate the underlying relationships between regional $\delta^{18}O_{precip}$ (and by extension $\delta^{18}O_{spel}$) and monsoon climate through the Holocene using multiple linear regression (MLR)"

From l286:

"We incorporate mean meteorological variables and $\delta^{18}O_{precip}$ for all Holocene time slices (1ka to 9ka) and all monsoon regions (CAM, ISM, EAM, SW-SAM, NE-SAM, SAfM, IAM) into the MLR model. Thus, the relationships constrained by the overall (global) MLR model represent the combined response across all monsoon regions."

From l349 (section 3.4):

"The global MLR model includes the Holocene (1 to 9ka) $\delta^{18}O_{precip}$ trends combined across all monsoon regions. This global monsoon MLR model has a pseudo-$R^2$ of 0.80…"

From l358:

"There are too few data points to make regressions for individual monsoon regions, but the distribution of data points for each region in the partial residual plots (Fig. 6) is indicative of the degree of conformity to the global MLR model (representing the combined response across all monsoon regions). Data points from the ISM, SW-SAM, IAM and SAfM are well aligned with the overall relationship with regional precipitation (Fig. 6a), indicating that precipitation is an important control on changes in $\delta^{18}O_{precip}$ in these regions. The NE-SAM, EAM and CAM values deviate somewhat from the overall relationship and, although there are relatively few points, this suggests that changes in precipitation are a less important influence on $\delta^{18}O_{precip}$ changes in these regions. The impact of temperature changes (Fig. 6b) in the ISM, EAM and SW-SAM is broadly consistent with the overall relationship. The slope of the relationship with temperature is negative for the IAM and NE-SAM, and since this is physically implausible it suggests that some factor not currently included in the MLR is influencing these records.  However, the inconsistencies between the regional signals helps to explain why the overall relationship between anomalies in temperature and $\delta^{18}O_{precip}$ is weak (Fig. 6b) and probably reflects the fact that tropical temperature changes during the Holocene are small. Data points from the EAM, ISM and IAM are well aligned with the overall relationship between changes in $\delta^{18}O_{precip}$ and changes in wind direction (Fig. 6c), indicating that changes in source area or moisture pathway are an important control on changes in $\delta^{18}O_{precip}$ in these regions. However, values for CAM, SW-SAM, NE-SAM and SAfM deviate strongly from the overall relationship."
* * *
Figure 6

Why are the predictor variables summer mean values whereas the d18Oprecip anomalies are precipitation weighted annual average values? Please clarify why summer mean d18Oprecip is not used.

The speleothem records used in this study are broadly-speaking in the monsoon domain and hence the signals will be dominated by the summer season rainfall. We will clarify this in the figure 6 caption:

"Figure 6: Partial residual plots from the multiple linear regression analysis, showing the relationship between anomalies in simulated $\delta^{18}O_{precip}$ and the four predictor variables, after taking account of the fitted partial effects of all the other predictors. The simulated $\delta^{18}O_{precip}$ are anomalies relative to the pre-industrial control simulation, and are annual values weighted by precipitation amount. The predictor variables are: precipitation in the delineated monsoon region (mm/d), temperature in the source region (°C), surface wind direction over the source region (°) as an index of potential changes in source region and the ratio of precipitation recycling to total precipitation over the monsoon region (RI, unitless). The predictor variables are summer mean values, representing the summer

monsoon, where summer is defined as May to September for northern hemisphere monsoons and November to March for southern hemisphere monsoons."
* * *
Discussion

"We have also shown that there is little difference in the isotopic values between the MH and the LIG in the ISM and EAM regions, which is also observed in individual speleothem records (Kathayat et al., 2016; Wang et al., 2008). Given that the increase in summer insolation is much larger during the LIG than the MH, this finding indicates that other factors play a role in modulating the monsoon response to insolation forcing and may reflect the importance of global constraints on the externally-forced expansion of the tropical circulation (Biasutti et al., 2018)."

What about greenhouse gasses as a possible explanation? Is the radiative forcing LIG to MH more similar compared to that of insolation forcing?

$CO_2$ concentrations are similar between the MH and LIG, based on ice core values (and as used as boundary conditions in the PMIP protocol: Otto-Bliesner et al., 2017). The MH and LIG have $CO_2$ concentrations of 264 and 275 ppm, respectively. Ice volumes may have also been slightly different between the two time periods: whilst MH ice volumes were similar to PI, higher sea-levels during the LIG (>6 m relative to today: Dutton and Lambeck, 2012) suggest lower ice volumes in Greenland and Antarctica at this time. The role of high-latitude ice volumes and $CO_2$ concentrations on East Asian monsoon variability has been widely discussed in loess record studies. Several studies show a strong correlation between EASM variability (as recorded by loess) and ice volumes/$CO_2$, with higher ice volumes and lower $CO_2$ driving a weaker monsoon (and vice versa), either due to a shift in the mean position of the ITCZ (Lu et al., 2013), or due to a strengthening and southward shift of the westerlies (Sun et al., 2015). On this basis, the slightly higher $CO_2$ and lower ice volumes of the LIG relative  to the MH would drive a stronger monsoon, reinforcing the summer insolation driven strengthening of the monsoon during the LIG. Thus, neither $CO_2$ or ice volume explain the similarity of the MH and LIG $\delta^{18}O_{spel}$ signals. It is, however, also worth noting that the monsoon front (where the EASM loess records are located) are likely more sensitive to high latitude ice-volume/$CO_2$ conditions, than the core of the EASM (Sun et al., 2015), where all the LIG speleothem records are located.

We will amend the text to include a brief discussion of $CO_2$/ice volume conditions of the MH vs. LIG signals.

From l411:

"We have also shown that there is little difference in the isotopic values between the MH and the LIG in the ISM and EAM regions, which is also observed in individual speleothem records (Kathayat et al., 2016; Wang et al., 2008). The LIG (125ka) period was defined by higher summer insolation, higher $CO_2$ concentrations (Otto-Bliesner et al., 2017) and lower ice volumes (Dutton and Lambeck, 2012) than the MH, suggesting that the LIG ISM and EAM monsoons should be stronger than the MH monsoons. The lack of a clear differentiation in the isotope signals between the LIG and MH suggests that other factors play a role in modulating the monsoon response to these forcings and may reflect the importance of global constraints on the externally-forced expansion of the tropical circulation (Biasutti et al., 2018)."
* * *
References:

Dutton, A. and Lambeck, K.: Ice volume and sea level during the last interglacial, Science, 337(6091), 216-219, 2012.

Fleitmann, D., Burns, S.J., Mudelsee, M., Neff, U., Kramers, J., Mangini, A., and Matter, A.: Holocene Forcing of the Indian Monsoon Recorded in a Stalagmite from Southern Oman, Science, 300(5626), 1737-1739, https://doi.org/ 10.1126/science.1083130, 2003.

Lu, H., Yi, S., Liu, Z., Mason, J.A., Jiang, D., Cheng, J., Stevens, T., Xu, Z., Zhang, E., Jin, L. and Zhang, Z.: Variation of East Asian monsoon precipitation during the past 21 ky and potential $CO_2$ forcing, Geology, 41(9), 1023-1026, 2013.

Otto-Bliesner, B.L., Braconnot, P., Harrison, S.P., Lunt, D.J., Abe-Ouchi, A., Albani, S., Bartlein, P.J., Capron, E., Carlson, A.E., Dutton, A. and Fischer, H.: The PMIP4 contribution to CMIP6–Part 2: Two interglacials, scientific objective and experimental design for Holocene and Last Interglacial simulations, Geosci. Model Dev., 10(11), 3979-4003, 2017.

Sun, Y., Kutzbach, J., An, Z., Clemens, S., Liu, Z., Liu, W., Liu, X., Shi, Z., Zheng, W., Liang, L. and Yan, Y.: Astronomical and glacial forcing of East Asian summer monsoon variability, Quat. Sci. Rev., 115, 132-142, 2015.

Weedon, G. P., Balsamo, G., Bellouin, N., Gomes, S., Best, M. J., and Viterbo, P.: The WFDEI meteorological forcing data set: WATCH Forcing Data methodology applied to ERA-Interim reanalysis data, Water Resources Research, 50, 7505–7514, https://doi.org/10.1002/2014WR015638, 2014.